# Phytochemical profiling of *Vitex negundo* seeds via UHPLC-QTOF-MS/MS analyses with antimicrobial evaluation and in silico targeting of DNA Gyrase B and Secreted Aspartic Proteinase 2 (SAP2)

Javed Mustafa[1], Tuba Ashraf[1], Basharat Ali[1], Shazia Kousar[1], Adeem Mahmood[2], Saif Ullah[3], Muhammad Imran[4]\*, Bakhat Ali[1,5]\*, Usman Rahim[1], Muhammad Yunis[1]

1 Institute of Chemistry, Khwaja Fareed University of Engineering and Information Technology, Rahim Yar Khan, Pakistan, 2 John Innes Centre, Norwich Research Park, United Kingdom, 3 Department of Chemistry, Govt Sadiq Abbas Graduate College, Dera Nawab Sahib, Bahawalpur, Pakistan, 4 Department of Chemistry, College of Science, King Khalid University, Abha, Saudi Arabia, 5 International Center for Chemical and Biological Sciences, H.E.J. Research Institute of Chemistry, University of Karachi, Karachi, Pakistan

\* bakhat.ali@kfueit.edu.pk (BA); imranchemist@gmail.com (MI)

## Abstract

The present study reports the metabolic profiling and antimicrobial evaluation of *Vitex negundo* seed extract. UHPLC-QTOF-MS/MS analysis identified seventeen bioactive phytoconstituents, correlated with the observed antimicrobial and antifungal activities. Among them, isoorientin ($-7.3$ kcal mol$^{-1}$), quercetin ($-7.8$ kcal mol$^{-1}$), and orientin ($-7.4$ kcal mol$^{-1}$) exhibited strong binding affinities towards Staph Gyrase B (24 kDa). Similarly, isoorientin ($-8.1$ kcal mol$^{-1}$), quercetin ($-8.4$ kcal mol$^{-1}$), and orientin ($-8.3$ kcal mol$^{-1}$) displayed significant interactions with secreted aspartic proteinase (SAP2) enzyme, confirming their antimicrobial potential. The aqueous-methanolic seed extract demonstrated notable inhibitory activity against *Staphylococcus aureus* ($26.4 \pm 0.3$ mm; 44.08% inhibition) and *Candida albicans* ($25.7 \pm 0.4$ mm; 29.73% inhibition). Density functional theory (DFT) calculations at B3LYP/6-31G level were used to optimize the ground state geometries of the identified phytochemicals and analyze their frontier molecular orbitals (FMOs) and global reactivity descriptors. Time-dependent DFT (TDDFT) calculations at the B3LYP/6-311G level (solvent: DMSO) further explored their biological relevance and nonlinear optical (NLO) properties, including ionization potential (IP), molecular electrostatic potential (MEP), and HOMO-LUMO energy gaps. These quantum chemical parameters provided mechanistic insights into the antimicrobial potential of the identified constituents. Molecular docking simulations further confirmed strong geometric complementarity and favorable binding affinities, highlighting the *Vitex negundo* seed extract as a promising source of a novel medicinal agent with previously unreported antifungal and antibacterial activities.

**Data availability statement:** All relevant data are within the paper and its Supporting Information files.

**Funding:** The authors extend their appreciation to the Deanship of Scientific Research at King Khalid University, Saudi Arabia, for funding this work through the Research Groups Program under Grant No. RGP.2/527/46. The authors are also deeply grateful to Dr. Muhammad Imran for his role as the project supervisor and lead researcher, and for his support in funding the study.

**Competing interests:** The authors have declared that no competing interests exist.

## Introduction

Herbal plants represent the primitive source of almost 25% of modern pharmaceuticals and have been employed for over 5000 years as pharmaceutical agents in the treatment of diverse ailments [1]. *Vitex negundo* L. (commonly known as Nirgundi or the five-leaved chaste tree) is a shrub of the family Verbenaceae, typically reaching a height of 2–5 m. The species is widely distributed across South Asia, including India, the Philippines, Pakistan, Afghanistan, Sri Lanka, and China [2] and has long been integrated into traditional medicine systems for decades [3]. In Unani and Ayurvedic practices, various parts of *V. negundo* have been extensively used for the management of numerous health disorders associated with human diseases [4,5]. Traditionally, *Vitex negundo* has been employed in the treatment of various microbiological infections, including tuberculosis (TB) [6], cholera, histoplasmosis, typhoid, pneumonia [7], urinary tract infections (UTI), influenza, COVID-19, hepatitis, ringworm, and aspergillosis [8]. Nearly all parts of the *Vitex negundo* are recognized for their medicinal value and are widely employed in pharmaceutical formulations [9,10]. The growing global concern over antimicrobial resistance has intensified research into alternative therapeutic agents. In this context, extensive investigations have highlighted the potential of *Vitex negundo* seeds, which exhibit notable biological activity against Gram-positive and Gram-negative bacteria as well as fungal pathogens [11]. The seeds are an integral component of traditional medicine and are regarded as a valuable source of bioactive compounds [12,13]. The seeds exhibit analgesic [14], anti-inflammatory [15], antioxidant [16], and diuretic [17] properties, primarily attributed to a diverse range of bioactive constituents, including terpenoids, lignans, steroids, and flavonoids [18].

Moreover, medicinal plants from diverse climatic and ecological regions have been proposed as a promising source of novel structural templates for antimicrobial drug discovery. These phytochemical frameworks may serve as the basis for developing new bioactive compounds to address the growing global challenge of microbial resistance to existing therapeutics [19,20]. Notably, nearly 80% of human microbial infections are associated with biofilm formation, which contributes to persistent and recurrent infections that often exhibit resistance to conventional treatments. [21]. The biofilm formation can protect fungal pathogens from the host's adaptive immune system. By impairing host immune responses, the biofilm environment diminishes the overall efficacy of antifungal therapies. The predominant causative agents of such infections are *Candida* and *Aspergillus* species [22,23]. To mitigate these infections, numerous medicinal plant species, particularly those inhabiting semi-arid and arid regions, have demonstrated significant antimicrobial potential [24]. Their phytoconstituents exhibit sustainable antimicrobial efficacy, with structural diversity contributing to their broad-spectrum therapeutic potential. Secondary metabolites such as flavonoids, saponins, phenolics, and triterpenes, along with nitrogen-containing phytochemicals, are of particular interest due to their diverse biological activities. Among these, alkaloids have recently gained considerable attention for their potent antimicrobial properties [25]. However, significant challenges still exist in identifying

effective plant-derived antimicrobials, and continued efforts are focused on discovering novel molecular templates from diverse ecological regions.

The implementation of traditional practices and knowledge related to medicinal plants has stimulated the global interest in their therapeutic potential for managing chronic diseases [26]. Medicinal plants play a vital role in modern healthcare, contributing nearly 25% of the pharmaceuticals currently used in developed countries. Such pharmacological drugs are being employed in the treatment of diverse infectious diseases in humans. Their enduring appeal stems from a long history of therapeutic application, particularly within the traditional medicine system of developing countries [27]. Among their diverse phytoconstituents, flavonoids exhibit notable biochemical properties that support pharmacological activity and provide protection against oxidative stress [28]. Density functional theory (DFT) serves as a powerful tool for evaluating biological potential, quantitative structure–activity relationships (QSAR), active sites, and molecular interactions of bioactive compounds. To elucidate the pharmacological relevance of the identified molecules, parameters such as molecular electrostatic potential (MEP), ionization potential (IP), and frontier molecular orbitals (FMOs) were computed. These quantum chemical evaluations provide valuable insights into molecular reactivity and interaction behavior. Our research group has previously reported several phytochemical and DFT-based investigations [29–32].

The present study aimed to establish a comprehensive scientific framework for the identification and characterization of new bioactive constituents from *Vitex negundo* seeds and to evaluate their biological significance. For the first time, major compounds and chemical classes from the seed extract were identified and characterized using UHPLC-QTOF-MS/MS, while the antimicrobial potential of the methanolic extract and its fractions was systematically assessed. The in vitro antimicrobial results were further substantiated by ligand–receptor interactions to evaluate binding affinities and inhibitory potential against key antimicrobial targets. Furthermore, virtual screening of seed-derived phytoconstituents revealed promising chemical scaffolds, highlighting their antimicrobial efficacy and potential relevance in phytomedicine.

## Materials and methods

### Chemicals and reagents

All the reagents and chemicals used in this study were purchased from an analytical-grade research supplier. These included methanol ($CH_3OH$), ethanol ($CH_3CH_2OH$), sulfuric acid ($H_2SO_4$), anhydrous sodium carbonate ($Na_2CO_3$), sodium hydroxide (NaOH), and deionized water (Milli-Q system). The microbial strains *Staphylococcus aureus* (bacterial strain) and *Candida albicans* (fungal strain) were obtained from Merck and Sigma-Aldrich. Furthermore, reference standards of ciprofloxacin and amphotericin were also purchased from standard suppliers.

### Sample collection and authentication

Seeds of *Vitex negundo* were collected from the Choulistan Desert, District Rahim Yar Khan (Punjab, Pakistan). The plant material was authenticated and identified by a taxonomist at the Department of Botany, Khwaja Fareed University of Engineering and Information Technology (KFUEIT), Rahim Yar Khan, Pakistan. A voucher specimen number (KFUEIT/CHEM/19/2022) has been deposited in the departmental herbarium for future reference. To preserve the phytochemicals' integrity, the seeds were thoroughly washed and air-dried, quenched with liquid $N_2$, and stored at low freezing temperatures for two days. The dried material of seeds was subsequently ground into a fine powder, passed through a 60-mesh sieve, and preserved in a Ziplock bag until further use.

### Extract formation

The cleaned, dried, and finely-powdered seed material was macerated in methanol (MeOH) for one week. The extract was filtered and concentrated using a rotary evaporator at reduced pressure to obtain the crude methanolic extract. This extraction process was repeated three times to ensure the maximum recovery of metabolites. The combined filtrate

was concentrated, and the resulting fractions were concentrated and stored in Eppendorf tubes at 4 °C until further characterization.

## Antimicrobial assay

Microbiological culture-forming microbes were employed, including fungal strains such as *Candida albicans* and Gram-positive bacterial strains, *Staphylococcus aureus*. The antimicrobial activity of *Vitex negundo* seed extract was evaluated using the agar well diffusion method against Gram-positive bacteria (*Staphylococcus aureus*) and fungus (*Candida albicans*) [33–35]. Amphotericin B (10 μg/ml) and ciprofloxacin (10 μg/ml) were used as positive controls for antifungal and antibacterial activity, respectively. DMSO (dimethyl sulfoxide) served as a negative control. All assays were performed in biological triplicates, and results were expressed as mean zone of inhibition diameters (mm) ± standard deviation (SD) to confirm statistical reliability. The microbial inocula were standardized to the McFarland standard ($1 \times 10^6$ CFU mL$^{-1}$) and uniformly spread onto each dextrose agar well diffusion plate for fungal and bacterial assays, respectively. The *Vitex negundo* seed extract solution of 100 μL was filled in each well (6 mm in diameter) at concentrations of 0.15 and 0.30 mg mL$^{-1}$. The agar well diffusion plates were incubated for 24 hours at 37 °C for *S. aureus* and for 24 hours at 28 °C for *C. albicans* [36]. The detailed experimentation procedure is provided in Supporting Information [S1, S2] in S1 File.

## Docking studies

Molecular docking simulations were conducted for selected phytoconstituents (**1**, **4**, **5**, **11**, **12**, **14**, **15**, **16**, and **17**; Table 1) to optimize protein-ligand interactions, thereby elucidating the binding affinities of *Vitex negundo* seed constituents with fungal and bacterial target proteins. Docking studies were performed using AutoDock Vina (version 1.5.7), following standard preparation protocols that involved removal of heteroatoms and crystallographic water molecules, addition of polar

**Table 1. The identified compounds in the methanolic extraction of *Vitex negundo* seeds by the UHPLC-QTOF-MS/MS technique.**

| No. | Rt (min.) | Peak area | Peak height | m/z value (mass) | Molecular formula | Identified compounds | Fragment ions (m/z) of identified compounds |
|---|---|---|---|---|---|---|---|
| 1 | 2.44 | 17470 | 1471 | 168.05 | $C_8H_8O_4$ | Vanillic acid | 140, 122, 95, 80, 66, 42 |
| 2 | 4.06 | 29710 | 4723 | 197.064 | $C_{12}H_{20}O_2$ | Linalyl acetate | 151, 125, 123, 110, 93, 83, 77, 67, 45 |
| 3 | 3.06 | 19390 | 2217 | 171.05 | $C_{10}H_{18}O_2$ | cis-Linalool oxide | 93, 66, 55, 39 |
| 4 | 5.39 | 39510 | 553.7 | 448.192 | $C_{21}H_{20}O_{11}$ | Isoorientin | 150, 147, 85 |
| 5 | 4.06 | 29710 | 4723 | 197.064 | $C_{12}H_{20}O_2$ | Geranyl acetate | 151, 125, 123, 110, 93, 83, 73, 67, 45 |
| 6 | 1.86 | 54740 | 41110 | 132.034 | $C_9H_8O$ | Cinnamaldehyde | 66, 61, 46, 45, 44, 41, 39, 34 |
| 7 | 5.19 | 23240 | 2312 | 229.043 | $C_{14}H_{28}O_2$ | Myristic acid | 211, 183, 155, 128, 101, 43 |
| 8 | 2.21 | 48450 | 3886 | 173.046 | $C_{10}H_{20}O_2$ | Decanoic acid | 114, 113, 72, 43 |
| 9 | 5.14 | 30420 | 3417 | 285.113 | $C_{18}H_{36}O_2$ | Stearic acid | 285, 85 |
| 10 | 9.09 | 42460 | 34620 | 436.316 | $C_{31}H_{64}$ | Hentriacontane | 167, 149, 71, 46 |
| 11 | 2.11 | 56340 | 4875 | 440.046 | $C_{30}H_{62}$ | Triacontane | 422, 176, 163, 145, 127, 85, 97, 91, 69 |
| 12 | 1.51 | 51220 | 10970 | 408.075 | $C_{29}H_{60}$ | Nonacosane | 229, 174, 163, 145, 97, 85, 71, 69 |
| 13 | 3.53 | 35310 | 4474 | 464.15 | $C_{33}H_{68}$ | Tritriacontane | 318, 271, 249, 207, 145, 127, 97, 85, 81 |
| 14 | 2.33 | 26930 | 1719 | 278.129 | $C_{18}H_{30}O_2$ | Linolenic acid | 157, 133, 60 |
| 15 | 5.27 | 19620 | 2575 | 322.106 | $C_{19}H_{30}O_4$ | Vitedoin B | 304, 245, 217, 188, 157, 140, 134, 84, 58 |
| 16 | 5.11 | 71760 | 7336 | 303.029 | $C_{15}H_{10}O_7$ | Quercetin | 257, 229, 198, 170, 153, 85, 64 |
| 17 | 5.39 | 39510 | 553.7 | 448.192 | $C_{21}H_{20}O_{11}$ | Orientin | 448, 147, 185 |

The relative percentages of all the identified compounds were calculated from the methanolic plant extracts based on the total peak area in the ion chromatogram.

hydrogens, and assignment of Kollman partial charges. Structural visualization and interaction analysis were carried out using Discovery Studio Visualizer (DSV 2024) [37,38]. The active site coordinates of Staph Gyrase B (24 kDa) were set at $x = -1.42$, $y = 0.322$, $z = -13.056$, with a grid box size of $40 \times 40 \times 40$ Å and a spacing of 1.0 Å. For secreted aspartic proteinase (SAP2), the grid center was defined at $x = 41.695$, $y = 24.906$, $z = 13.627$, using a grid box of $30 \times 30 \times 30$ Å and the same spacing. Docking validation was performed by re-docking the co-crystallized ligand into its respective binding site, yielding RMSD values ≤ 2.0 Å, confirming the reliability and reproducibility of the docking protocol. To ensure comparative accuracy, control docking was conducted using *Staphylococcus aureus* Gyrase B (24 kDa, PDB ID: 4URO) and *Candida albicans* secreted aspartic proteinase (SAP2) (PDB ID: 1EAG). Ciprofloxacin (–7.8 kcal mol⁻¹) and amphotericin B (–10.0 kcal mol⁻¹) served as reference antibacterial and antifungal standards, respectively, for benchmarking the binding affinities of the seed-derived phytoconstituents. Docking simulations assessed the interaction strength, spatial orientation, and geometric complementarity of the ligands within the active sites, with binding energies (kcal mol⁻¹) used as the principal evaluation parameter. The 3D crystal structures of target proteins were retrieved from the Protein Data Bank (PDB), and the 3D ligand structures were obtained from the PubChem database in PDB format. Before docking, all crystallographic water molecules, heteroatoms, and cofactors were removed from the protein structures [39]. Each ligand was docked alongside its respective reference drug to ensure procedural consistency. Docking was performed using AutoDock Vina, generating nine conformations for each compound. The best binding poses were selected based on hydrogen bonding, π–π stacking, and hydrophobic interactions with key active-site residues. Protein–ligand interactions were further analyzed according to binding-site accuracy, interacting amino acids, hydrogen bond formation, and docking energy values [40].

## Computational studies

First-principles calculations provide valuable insight into diverse characteristics of bioactive constituents. Density functional theory (DFT) has demonstrated a reliable method for exploring the electronic structures of materials and reproducing experimental observations [41]. It has also been extensively employed to optimize ground-state (S₀) geometries of bioactive compounds [42]. Density Functional Theory (DFT) calculations were achieved by using Gaussian 09 to observe the electronic parameters of the most active top-scoring constituents. Among the available functionals, B3LYP is recognized as one of the most balanced and widely employed theories [43]. In the present study, ground-state optimization was carried out using the B3LYP/6-31G and Triple-zeta polarization (TZP) basis set as the Amsterdam-Density-Functional (ADF) package [41b]. TDDFT was employed at B3LYP/6-311G level (solvent DMSO) [44]

## Results and discussion

### UHPLC-QTOF-MS/MS-based metabolic profiling of *Vitex negundo* seeds under positive ionization mode

A non-targeted UHPLC-QTOF-MS/MS analysis of *Vitex negundo* seed extract was performed in triplicate to ensure retention time reproducibility, maintaining a relative standard deviation (RSD) below 0.2% across the replicate injections. Chromatographic separation was achieved by using a C18 reversed-phase column (2.1 × 100 mm, 1.7 µm) operated under a gradient elution program with a binary solvent system comprising solvent A (0.1% formic acid in water) and solvent B (0.1% formic acid in acetonitrile). The instrumental calibration was performed using authentic reference standards, including casticin, luteolin, and quercetin, with calibration curves exhibiting excellent linearity (R² > 0.995) among the tested concentration range (0.1–10 µg mL⁻¹). Extracted ion chromatograms (EICs) (Fig 1) were analyzed to assess possible co-elution effects among the 17 identified compounds (Fig 3), identifying distinct peak separation and confirming compound purity. The high mass accuracy (<5 ppm error) and consistency of retention times validated the accurate identification of *Vitex negundo* seed phytochemicals. The detailed metabolic procedure is provided in Supporting Information [S3] in S1 File. Compound (**1**) was eluted at a retention time of 2.44 min and identified as vanillic acid (Fig 2, Table 1). The compound exhibited a [M + H]⁺ ion at m/z 168, consistent with the molecular formula $C_8H_8O_4$. The mass spectrum displayed

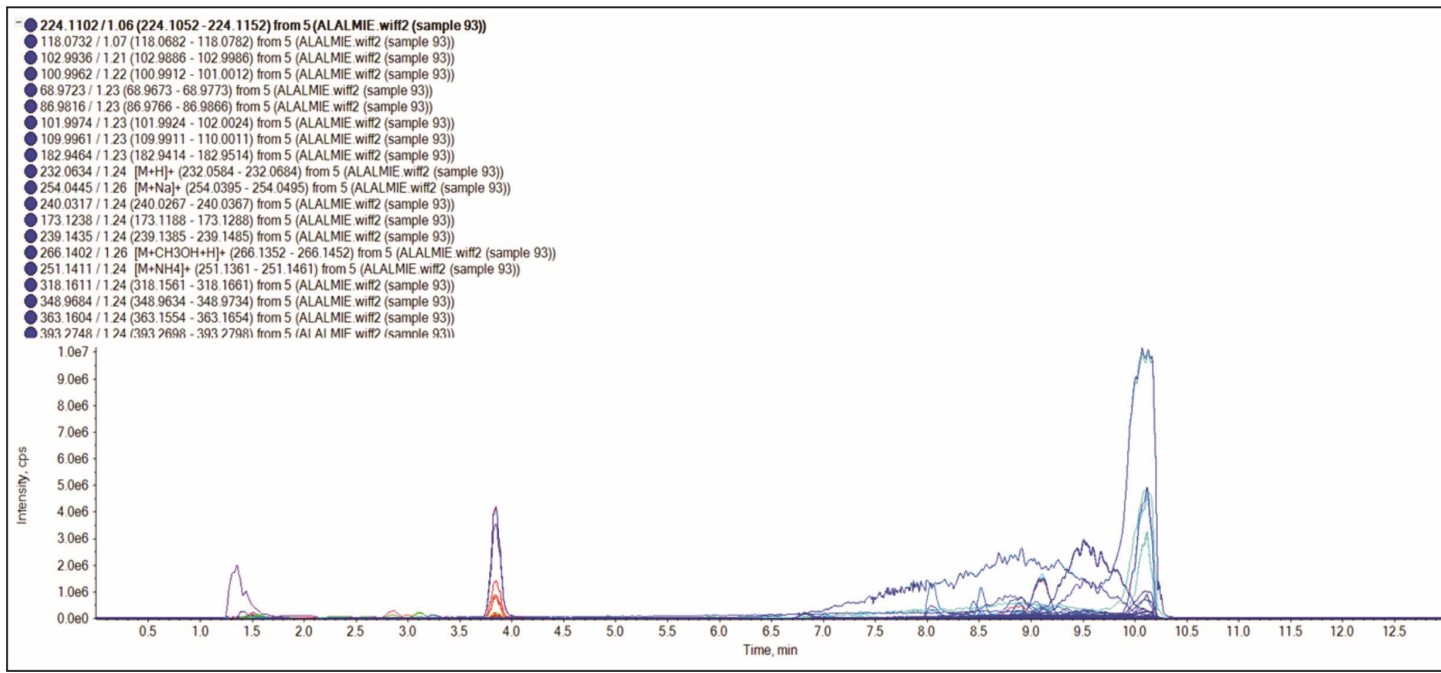

**Fig 1. UHPLC-QTOF-MS/MS extracted ion chromatogram of methanolic extraction of *Vitex negundo* seeds.**

characteristic fragment ions at m/z 122 [M+H-46]⁺, corresponding to the loss of a carboxyl group (COOH), and at m/z 95 [M+H-73]⁺, and m/z 42 [M+H-126]⁺, attributed to cleavage within the aromatic carbon skeleton. Vanillic acid, a phenolic acid, has been reported in the literature as a natural antioxidant, antimicrobial, antidiabetic, and anti-cancer agent [45]. Compound (**2**) produced a protonated molecular ion [M+H]⁺ at m/z 197 with a retention time (Rt) of 4.06 min. The MS/MS spectrum revealed fragment ions at m/z 151 [M+H-46]⁺, 125 [M+H-72]⁺ (loss of $O_2$ and $C_4H_7$· radical) and 77 [M+H-120]⁺, along with additional peaks at m/z 123, 93, and 67. Based on the fragmentation pattern, the molecular formula $C_{12}H_{20}O_2$ was assigned, and the compound was identified as linalyl acetate (Fig 2, Table 1) [46].

Compound (**3**) eluted at a retention time of 3.06 min, exhibiting a molecular protonated ion [M+H]⁺ at m/z 171, corresponding to molecular formula $C_{10}H_{18}O_2$. The MS/MS spectrum showed a fragment ion at m/z 68 [M+H-103]⁺, resulting from cleavage between C–O (loss of $C_5H_{10}O_2$·) and C–C bonds of the parent ion. Additional peaks appeared at m/z 93 [M+H–78]⁺ and m/z 55 [M+H–116]⁺ with high intensity. Based on fragmentation data, the compound was identified as cis-linalool oxide (Fig 2, Table 1), a known phytoconstituent of *Vitex negundo* seeds with reported antimicrobial activity [47]. Compound (**4**) eluted at a retention time of 5.39 min, exhibiting a protonated molecular ion [M+H]⁺ at m/z 448. The MS/MS spectrum showed a prominent fragment ion at m/z 147 [M+H-301]⁺, resulting from cleavage between two carbon atoms with elimination of $C_{15}H_9O_6$·. Another characteristic peak appeared at m/z 85 [M+H-363]⁺. Based on these fragment patterns and literature comparison, the compound was identified as isoorientin in Fig 2 and Table 1 [48]. Compound (**5**) eluted at a retention time of 4.06 min, producing a protonated molecular ion

[M+H]+ at m/z 197.064 in positive ionization mode, corresponding to a molecular mass of 196 and the expected molecular formula $C_{12}H_{20}O_2$. The dominant peak with high intensity was appeared at m/z 123 [M+H-74]+, arising from C–C bond cleavage to form the $C_9H_{15}$· fragment. Additional fragments ions appeared at m/z 151 [M+H-46]+, and 83 [M+H-114]+. Based on these fragmentation patterns and literature data, the compound was identified as geranyl acetate (Fig 2 and Table 1) [49]. Compound (**6**) was detected in positive ionization mode with a retention time of 1.86 min, showing a protonated molecular ion

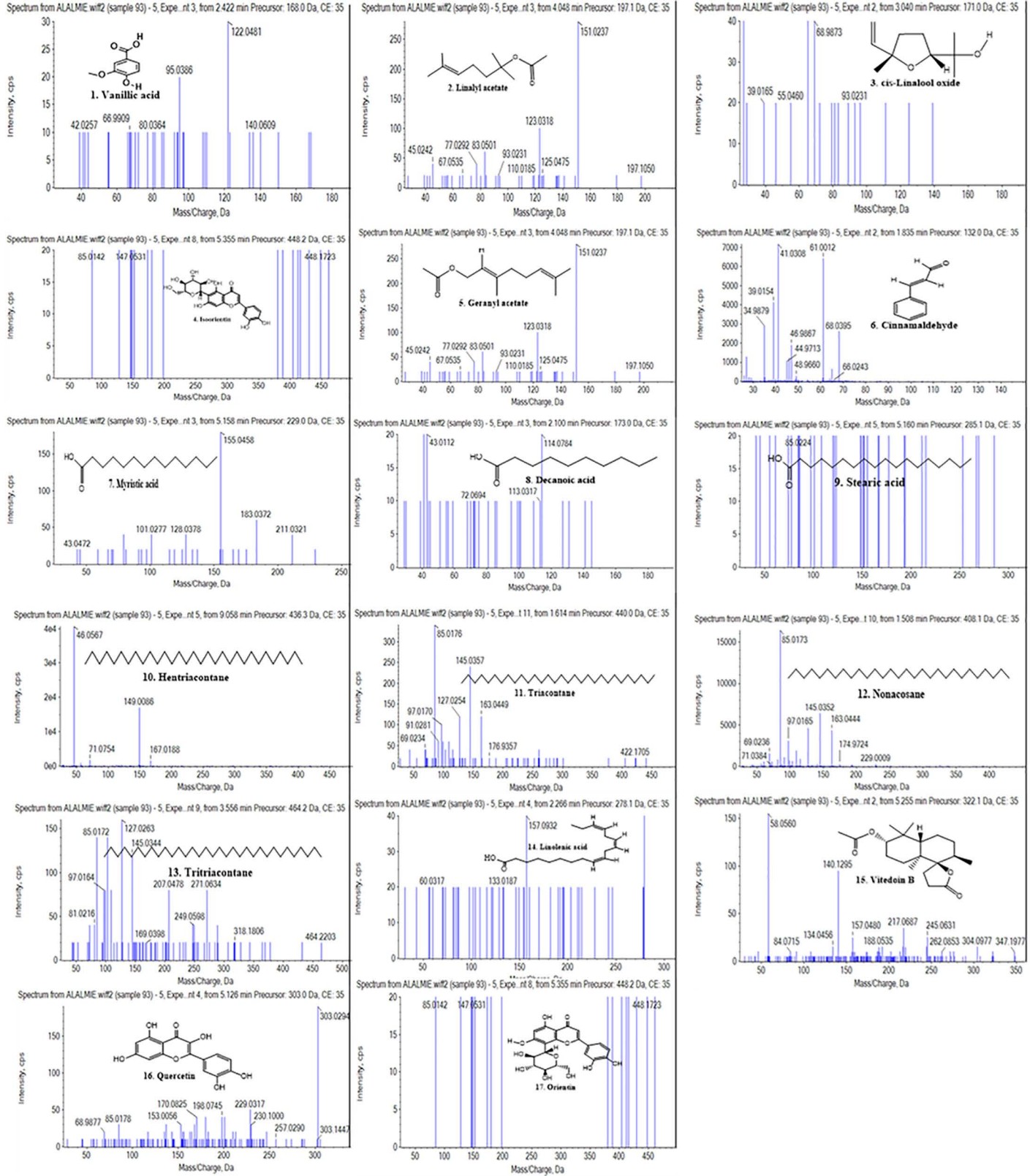

**Fig 2. MS/MS spectra and structures of compounds (1-17) identified in *Vitex negundo* seeds.**

[M+H]+ at m/z 132, with corresponding molecular formula $C_9H_8O$. The MS/MS spectrum displayed a dominant fragment at m/z 41 [M+H-91]+, resulting from C=C bond cleavage, and a secondary fragment at m/z 39 [M+H–93]+, attributed to benzene ring cleavage. An additional fragment was observed at m/z 39 [M+H-93]+. Based on fragmentation pattern and literature data, the compound was identified as cinnamaldehyde (Fig 2 and Table 1) previously reported in *Vitex negundo* seeds [50,51]. Compound (**7**) was detected in positive ionization mode [M+H]+, with a retention time of 5.19 min, exhibiting a protonated molecular ion [M+H]+ at m/z 229.043, consistent with the molecular formula $C_{14}H_{28}O_2$. The present compound resulted in a prominent peak at 183 [M+H-46]+ formed by the corresponding loss of carboxylic acid (COOH) group, and additional peaks at m/z 155 [M+H-74]+ and 128 [M+H-101]+. Finally, the compound was detected as myristic acid and proved in Fig 2 and Table 1 [52]. Compound (**8**) was analyzed in positive ionization mode with protonated molecular ion [M+H]+ at m/z 173.046, corresponding to the molecular formula $C_{10}H_{20}O_2$ with the retention time of 2.21 min. The most dominant peak was identified at m/z 43 [M+H-130]+ by fragmentation between two carbon atoms. Other fragment peaks resulting from the MS/MS spectrum appeared at m/z 114 [M+H-59]+ and 72 [M+H-101]+. By comparing with previous literature and fragmentation patterns, the compound was identified as decanoic acid, as illustrated in Fig 2 and Table 1 [53]. Compound (**9**) was identified as stearic acid with an observed retention time of 5.14 min, characterized with the help of LC-MS/MS analysis displaying the molecular formula $C_{18}H_{36}O_2$ as revealed in the mass spectrum (Fig 2 and Table 1). The precursor mass to the charge ion was observed at 285.113 in the positive mode of ionization. The MS/MS spectrum showed a prominent fragment at m/z 85 [M+H-200]+ due to cleavage between two carbon atoms with loss of radical $C_6H_{13}$˙. Based on the fragmentation pattern and literature data, the compound was identified as stearic acid. Stearic acid is a saturated monounsaturated fatty acid found in *Vitex negundo* seeds extract [54]. Compound (**10**) was identified as hentriacontane (Fig 2 and Table 1) with a retention time of 9.09 min. In positive ionization mode, the compound exhibited a protonated molecular ion [M+H]+ at m/z 436.316, corresponding to the molecular formula $C_{31}H_{64}$. The molecular mass of a compound was detected at 436. The compound displayed its most intense peak at m/z 71 [M+H-365]+, corresponding to cleavage between two carbon atoms by loss of radical $C_{26}H_{53}$˙. Additional fragment ions of the bioactive compound were observed at m/z 46 [M+H-390]+ and 149 [M+H-287]+, respectively. Hentriacontane is a long-chain alkane hydrocarbon reported for its vermifuge activity [55] (Fig 3).

Compound (**11**) was characterized with a retention time of 2.11 min, consisting molecular formula of $C_{30}H_{62}$ and identified as triacontane (Supporting Information S4 in S1 File) [56]. The Compound (**12**), eluting at a retention time of 1.51 min, was assigned the molecular formula of $C_{29}H_{60}$ and identified as nonacosane (Table 1; Fig 2, Supporting Information S5 in S1 File) [57]. Compound (**13**), with a retention time of 3.53 min, was characterized as tritriacontane, consisting molecular formula of $C_{33}H_{68}$ (Table 1; Fig 2, Supporting Information S6 in S1 File) [58]. The Compound (**14**), eluting at Rt=2.33 min, and corresponding to the molecular formula of $C_{18}H_{30}O_2$, was identified as linolenic acid (Supporting Information S7 in S1 File) [59]. The Compound (**15**) eluted at a retention time of 5.27 min, was identified as vitedoin B with consisting molecular formula of $C_{19}H_{30}O_4$ (Supporting Information S8 in S1 File) [51b]. The Compound (**16**) detected with its retention time of 5.11 min and was exposed as quercetin in positive ionization mode, representing a molecular formula of $C_{15}H_{10}O_7$ as shown in Fig 2 and Table 1. The protonated molecular ion [M+H]+ was observed at m/z 303.029. The molecular weight of quercetin was found to be 302. The most prominent fragment appeared at m/z 153 [M+H-149]+, resulting from the benzene ring cleavage. Additional fragments were detected at m/z 170 [M+H-132]+, 198 [M+H-104]+, and 230 [M+H-72]+. Quercetin, a flavonoid polyphenol, has been previously reported in *Vitex negundo* seeds [60]. The Compound (**17**) was identified at retention time (Rt) of 5.39 min in positive ionization mode. The protonated molecular ion [M+H]+ was detected at m/z 448.192, corresponding to the molecular formula of $C_{21}H_{20}O_{11}$. The most prominent fragment appeared at m/z 147 [M+H-301]+, after cleavage between two benzene rings, and the elimination of fragments, $C_6H_{11}O_5$˙ and $C_3H_5O_3$²˙along with proton [H+]. An additional high-intensity fragment was observed at m/z 85 [M+H-363]+ in the MS/MS spectrum. By comparing with previous literature and fragmentation pattern, the compound was identified as orientin, as shown in Fig 2 and Table 1 [61].

In the present study, seventeen phytoconstituents were identified in the methanolic seed extract of *V. negundo* through UHPLC-QTOF-MS/MS analysis. Several of these compounds possess significant therapeutic potential

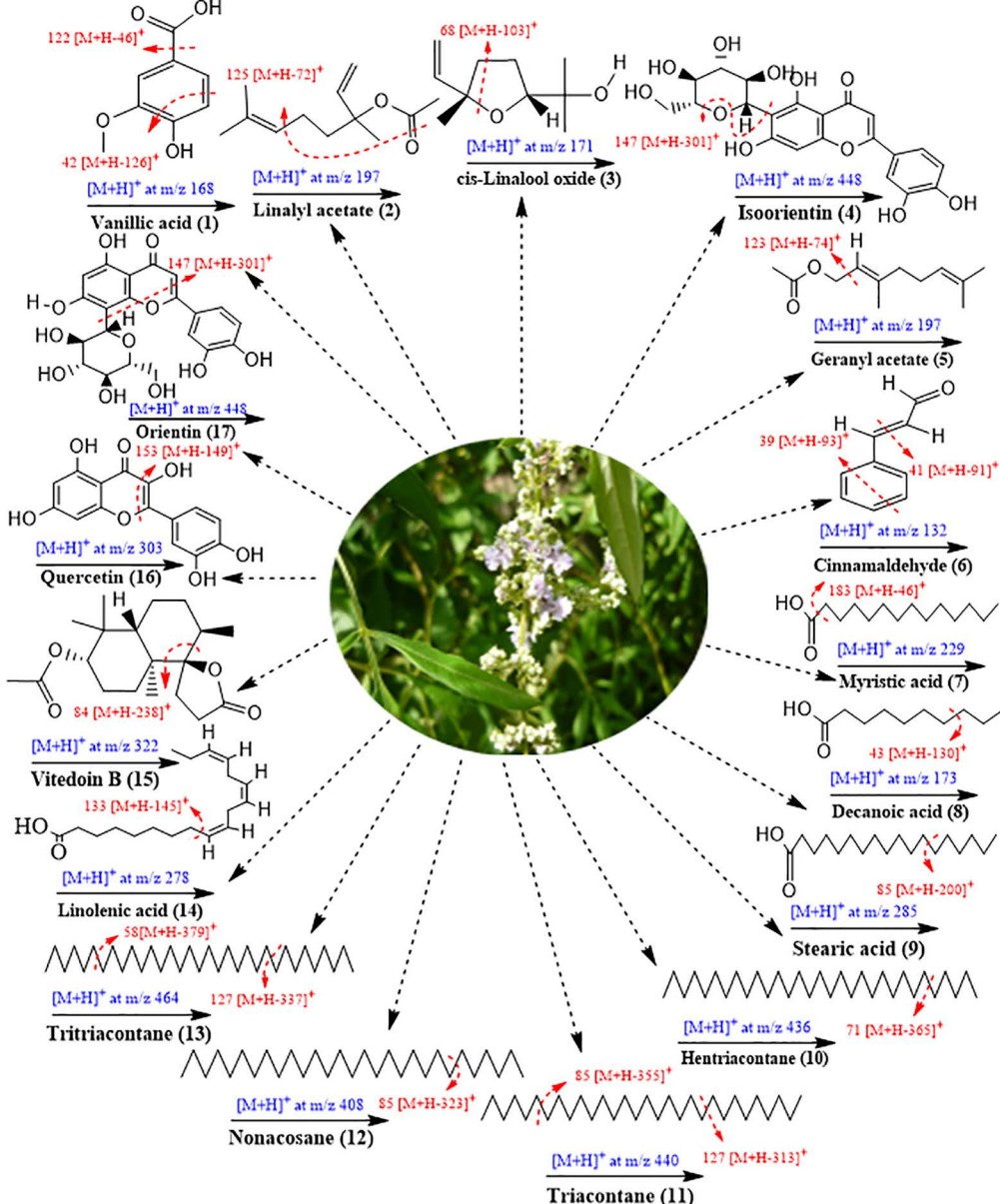

**Fig 3. Proposed fragmentation scheme of identified metabolites (1-17) from the methanolic extract of *Vitex negundo* seeds.**

encompassing a broad spectrum of bioactive constituents. The synergistic interplay between their antimicrobial and antioxidant mechanism suggests that *Vitex negundo* seed extract may serve as a promising natural source for functional food and nutraceutical applications, particularly in mitigating microbial infections, oxidative stress, and inflammation-related diseases.

## The anti-microbial biofilm activities of the MeOH extract of *Vitex negundo* seeds

The antimicrobial activity of the crude methanolic extract of *Vitex negundo* seeds [62] was assessed in vitro against a Gram-positive bacterium, *Staphylococcus aureus,* and the fungal strain *Candida albicans*, using ciprofloxacin and

amphotericin as respective reference standards (Table 2). The antimicrobial efficacy was determined by measuring inhibition zones (mm), which indicate the antimicrobial efficiency of the extract compared with standard drugs. The results are shown in Fig 4. The seed extract produced a zone of inhibition 26.4±0.3 mm against *Staphylococcus aureus*, corresponding inhibition potency of 44.08% compared with ciprofloxacin (28.7±0.5 mm). In contrast, the antifungal activity against *Candida albicans* was lower, with inhibition zones of 25.7±0.4 mm compared to amphotericin (27.5±0.4 mm), representing 29.73% inhibition potency. All antimicrobial assays were performed in biological triplicates, and results were presented as mean standard deviation to enhance statistical reliability. The DMSO (negative control) produced no inhibition zone, indicating that antimicrobial efficiency was attributed solely to the seed extract. The bioactive constituents of *Vitex negundo* seed extract exhibited stronger antibacterial activity compared to antifungal activity. The higher inhibition potency of seed extract against *S. aureus* may be attributed to a greater affinity of the extract for bacterial targets such as intracellular enzymes or cell wall components; conversely, the lower efficacy against *Candida albicans* suggests reduced interactions with fungal cellular structures, likely due to differences in the structural and physiological barriers between fungal and bacterial cells [63].

$$\text{Zone of inhibition \%} = \text{Antilog}\left[\frac{(TH + TL) - (SH + SL)}{(TH - TL) - (SH - SL)} \times 0.30\right] \times 100$$

**Table 2. Anti-microbial activities of MeOH seeds extract of *Vitex negundo*.**

| Microorganism | Reference drug (10 μg/ml) | ZOI of reference drug (mm) | ZOI of seed extract (mm, mean±SD, n=3) | Seed extract concentration (mg/ml) |
|---|---|---|---|---|
| **Gram-positive bacteria** | | | | |
| *Staphylococcus aureus* | Ciprofloxacin | 28.7±0.5 | 26.4±0.3 | 0.30 |
| **Fungi** | | | | |
| *Candida albicans* | Amphotericin B | 27.5±0.4 | 25.7±0.4 | 0.30 |

All standard fractions and sample values are mentioned in mg/ml; ZOI: zone of inhibition measured in millimeters, SD: standard deviation, MeOH: methanol (solvent), DMSO: dimethyl sulfoxide.

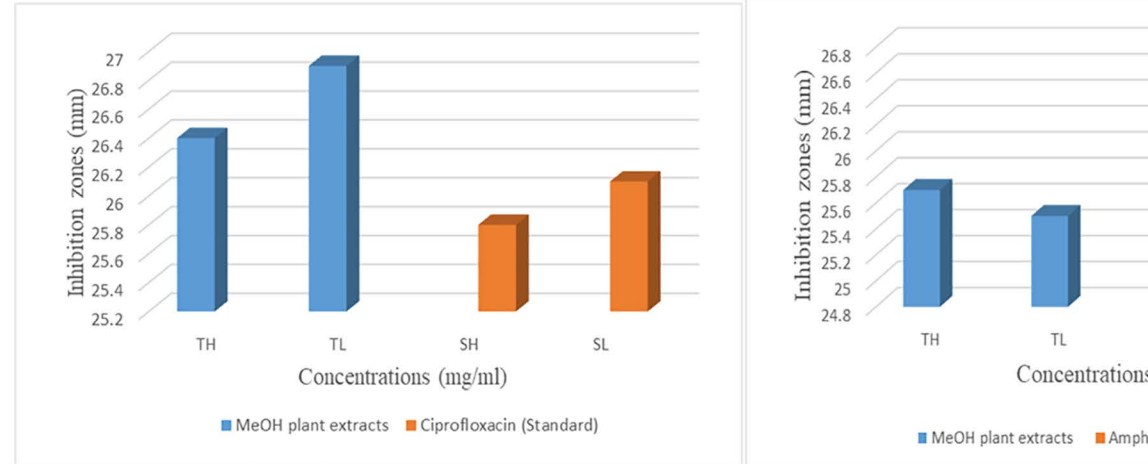

**Fig 4. Antimicrobial activity comparison of *Vitex negundo* seed extract with ciprofloxacin and amphotericin.**

The experimental antimicrobial activity was further supported by docking simulations and quantum chemical analyses. The major seed-derived flavonoids (phytochemicals) isoorientin, orientin, and quercetin exhibited strong binding affinities towards microbial enzymes, including *Staphylococcus aureus* DNA Gyrase B (24 kDa) and *Candida albicans* secreted aspartic proteinase (SAP2), with binding energies ranging from –7.3 to –8.4 kcal mol$^{-1}$, signifying strong inhibitory potential and geometric complementarity. Moreover, small HOMO–LUMO energy gaps and the molecular electrostatic potential (MEP) analyses revealed electron-rich regions on carbonyl and hydroxyl groups, which are responsible for strong hydrogen-bonding interactions.

The combined in silico and in vitro findings support the antimicrobial potential of flavonoid-rich phytochemicals in the seed extract of *Vitex negundo*. These findings provide a rational framework for the development of natural antimicrobial agents from *Vitex negundo*, enhancing their medicinal significance following computational validation.

## Docking simulations of major bioactive constituents as antibacterial and antifungal agents

For molecular docking studies, target bioactive compounds **1, 4, 5, 11, 12, 14, 15, 16,** and **17** (Table 1) were selected based on their structural classification, including phenolic acids, alkaloids, phytoalexins, and flavonoids [64]. The inhibitory potential of these selected plant-derived compounds was evaluated against *Staphylococcus aureus* Gyrase B (24 kDa) and secreted aspartic proteinase (SAP2), with ciprofloxacin and amphotericin used as antibacterial and antifungal drug standards, respectively. Among the tested compounds, isoorientin (**4**), quercetin (**16**), and orientin (**17**) exhibited improved antibacterial and antifungal activities. Moreover, these compounds determine an auspicious inhibitory potential associated with the reference standard, representing therapeutic applications for lead compounds. Isoorientin (**4**), quercetin (**16**), and orientin (**17**) displayed the nearest ΔG values of –7.3 kcal mole$^{-1}$, –7.8 kcal mole$^{-1}$, and –7.4 kcal mole$^{-1}$, respectively, compared with ciprofloxacin, ΔG –7.8 kcal mole$^{-1}$ against *S. aureus* Staph Gyrase B (Table 3).

In addition to antibacterial targets, the selected bioactive compounds also demonstrated strong inhibition effects against the antifungal reference amphotericin (ΔG –10.0 kcal mol$^{-1}$). The isoorientin (**4**), quercetin (**16**), and orientin (**17**) exhibited binding affinities of ΔG –8.1 kcal mol$^{-1}$, ΔG –8.4 kcal mol$^{-1}$, and ΔG –8.3 kcal mol$^{-1}$, respectively. The stronger inhibition effect was shown by the flavonoid constituent, i.e., quercetin (**16**), against the ciprofloxacin and amphotericin reference standards. The isoorientin (**4**) has displayed a well-defined interaction with macromolecules, forming three hydrogen bond interactions and four hydrophobic interactions (pi-pi, pi-anion, pi-alkyl, and pi-sigma) holding amino acid

**Table 3. *Vitex negundo* constituents in methanolic seed extract and their binding interaction energies (kcal mole$^{-1}$) with Staph Gyrase B 24 kDa, secreted aspartic proteinase (SAP2).**

| Compound Name and Entry # in Table 1 | Energy (kcal/mole) at 4URO | Energy (kcal/mole) at 1EAG |
|---|---|---|
| Vanillic acid (**1**) | –5.8 | –5.4 |
| Isoorientin (**4**) | –7.3 | –8.1 |
| Geranyl acetate (**5**) | –5.6 | –5.4 |
| Triacontane (**11**) | –6.1 | –6.7 |
| Nonacosane (**12**) | –6.0 | –6.3 |
| Linolenic acid (**14**) | –5.5 | –5.4 |
| Vitedoin B (**15**) | –6.3 | –7.3 |
| Quercetin (**16**) | –7.8 | –8.4 |
| Orientin (**17**) | –7.4 | –8.3 |
| Ciprofloxacin (standard antibacterial drug) | –7.8 | _ |
| Amphotericin (standard antifungal drug) | _ | –10.0 |

residues, i.e., Thr173, Gly85, Arg84, Glu58, Asp57, Asn54, and Ile86, respectively (Fig 5 and Table 3). The bond distance of these interactions ranged from 1.10212 to 4.7832 Å.

Isoorientin, when docked with macromolecule 4URO, exhibited the docking score of −7.3 kcal mole$^{-1}$. The ligand formed three conventional hydrogen bond interactions (HBIs) via its hydroxyl (OH$^-$) groups with the amino acid residues Asn54, Glu58, and Thr173 of the target protein. In addition, 2D interactions mapping revealed further stabilization by non-covalent interactions with residues Gly85, Arg84, Ile86, and Asp57. Amino acid residues (AARs) involved in the

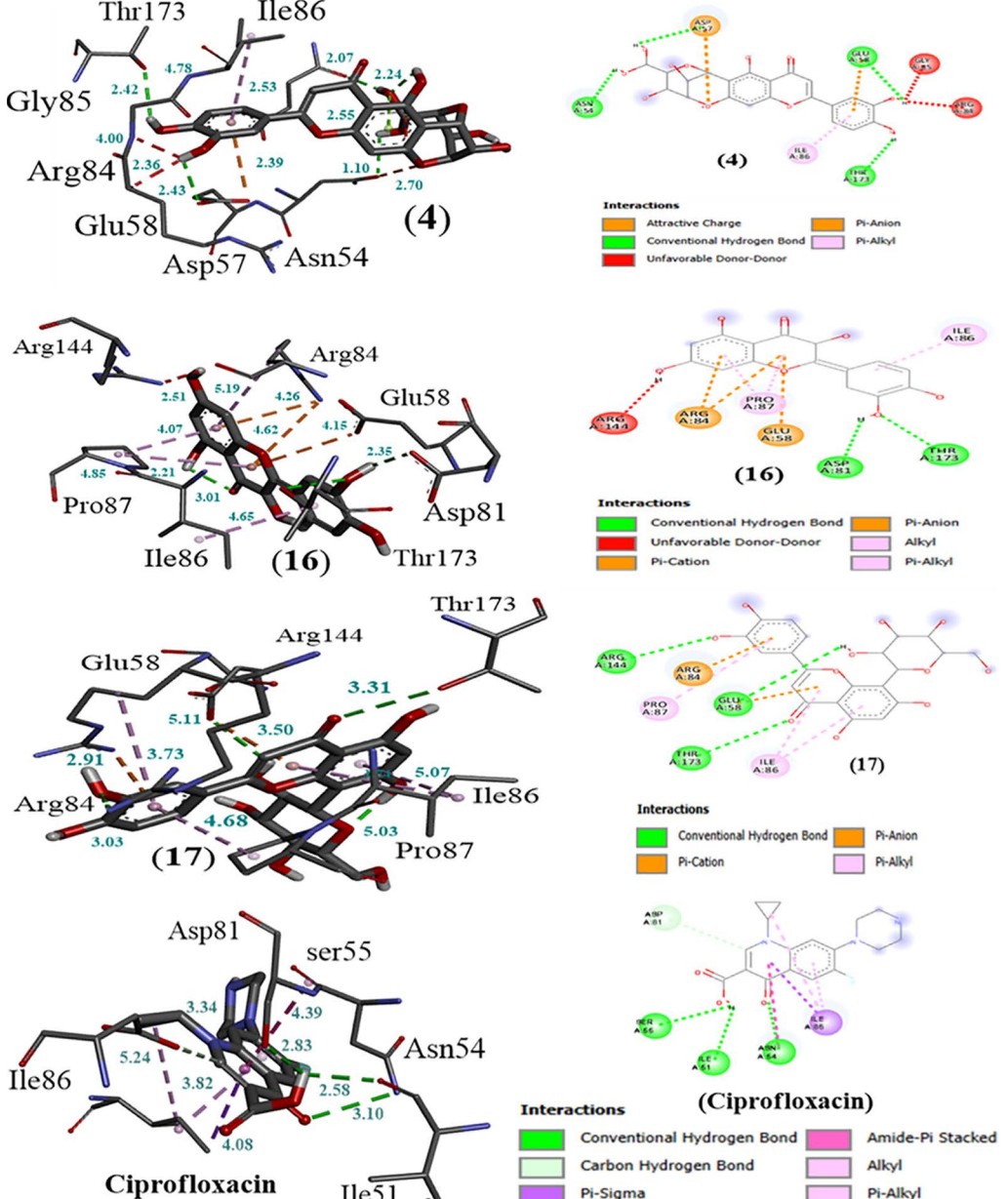

**Fig 5. The 3D and 2D interaction plot of identified bioactive compounds (4), (16), (17), and ciprofloxacin (standard) with Staph Gyrase B 24 kDa protein.**

binding of isoorientin included Asn54, Asp57, Glu58, Gly85, Arg84, Thr173, and Ile86 with corresponding bond lengths of 2.53, 2.70, 4.00, 2.36, 2.43, 2.55, and 4.78 Å, respectively (Fig 5 and Table 3).

Quercetin (**16**) has demonstrated enhanced antimicrobial potential through molecular docking, with the active binding sites of protein Staph Gyrase B (24 kDa; with PDB ID: 4URO), yielding a docking score of −7.8 kcal/mole. The phytochemical established two conventional hydrogen bonds and three hydrophobic interactions (pi-anion, pi-alkyl, and pi-cation) with amino acid residues Ile86, Thr173, Asp81, Glu85, Pro87, Arg84, and Arg144. The interaction bond lengths ranged from 2.21 to 5.19 Å (Fig 5 and Table 3). In this metabolite, two conventional hydrogen bonds were established between the hydroxy (OH⁻) group of the ligand and amino acid residues Thr173 and Asp81, with bond distances of 3.01 and 2.35 Å, respectively. Additionally, hydrophobic interactions were observed with Glu58 (4.15 Å), Arg84 (5.19 Å), Ile86 (4.85 Å), and Pro87 (4.07 Å). The phytochemical orientin (**17**) docked with *Staphylococcus aureus* DNA Gyrase B (PDB ID: 4URO), yielding a binding energy of −7.8 kcal mol⁻¹, along with an interaction involving Arg144. Additional hydrophobic interactions were observed among the molecules, including pi-anion, pi-cation, and pi-alkyl with amino acids like Ile86, Pro87, and Arg84, respectively, in a crystal structure of a protein. The bond distance of these interactions ranged between 2.91 to 5.11 Å (Fig 5 and Table 3). In the ligand-protein complex, specific amino acid residues such as Arg84 (3.50 Å), Glu58 (5.11 Å), Arg144 (3.03 Å), Thr173 (3.31 Å), Ile86 (5.07 Å), and Pro87 (1.74 Å) were engaged in stabilizing the binding. These interactions suggest that the phytochemical has exhibited strong antimicrobial potential relative to the reference standard ciprofloxacin when targeting *Staphylococcus aureus* DNA Gyrase B.

Isoorientin (**4**), a natural flavonoid, was also docked with protein secreted aspartic proteinase (SAP2, PDB ID: 1EAG) to evaluate its antifungal potential against the reference standard amphotericin. The compound exhibited inhibition values of −8.4 kcal mol⁻¹, which was comparable to that of amphotericin (−10.0 kcal mol⁻¹). The molecular binding interaction revealed four conventional hydrogen bonds with Asp86 (5.49 Å), Gly220 (3.19 Å), Gly34 (3.06 Å), and Glu193 (2.15 Å). Additional hydrophobic interactions were observed with amino acid residues Tyr84 (2.11 Å), Asp32 (2.26 Å), and Asp218 (2.87 Å). The isoorientin has shown a bond distance range with Sap2 interactions extended from 2.11 to 5.49Å (Fig 6 and Table 3). The docking result of isoorientin indicates a strong antifungal inclination, approaching that of the standard drug amphotericin.

The antifungal potential of quercetin (**16**) was further investigated by docking with the secreted aspartic proteinase (SAP2, PDB ID: 1EAG). The compound exhibited a binding affinity value of −8.4 kcal mol⁻¹, containing multiple binding interactions within the binding pocket. The interaction profile of the molecule included one conventional hydrogen bond with Asp32 and one alkyl interaction with amino acid Tyr84. Additional non-covalent interactions were observed with Gly87 (2.10 Å), Asp218 (5.60 Å), Ser88 (1.53 Å), Ile123 (2.13 Å), Leu124 (2.99 Å), and Ser35 (5.71 Å). The bond length displacement for such binding interactions varied from 1.5 Å to 5.71 Å (Fig 6 and Table 3).

Orientin (**17**) exhibited a binding affinity value of −8.3 kcal mole⁻¹, making a strong receptor-ligand complex by four hydrogen bonds (conventional hydrogen bonds) with two hydrophobic interactions (pi-pi) in the binding sphere within the active site of SAP2. The interacting amino acid residues included Thr222 (2.55 Å), Gly220 (2.46 Å), Asp32 (2.16 Å), Gly85 (5.21 Å), Asp86 (1.97 Å), and Tyr225 (4.63 Å), with overall bond distances ranging from 1.97 to 5.21 Å (Fig 6 and Table 3). These binding interactions of amino acid residues ranged from 1.97 to 5.21 Å. Based on docking evaluations, orientin possesses much binding potential, approaching the inhibitory efficiency of the reference antifungal drug amphotericin. The detailed binding interaction diagrams of promising compounds are provided in Supporting Information [S9] in S1 File.

The docking simulations demonstrated that *V. negundo* seed phytochemicals exhibited moderate to strong binding interactions with bacterial and fungal target proteins. The most active phytochemicals, such as isoorientin (**4**), quercetin (**16**), and orientin (**17**), displayed binding energies ranging from −7.3 and −8.4 kcal/mol, comparable to those of ciprofloxacin (−7.8 kcal/mol) and amphotericin (−10.0 kcal/mol). The control docking of amphotericin and ciprofloxacin validated the reliability of the docking scores, showing RMSD values ≤ 2.0 Å and hydrogen bonding interactions at the protein active sites with residues (Asn54, Arg84, Glu58, and Asp81). The reproducibility of the results was confirmed through consistency across triplicate

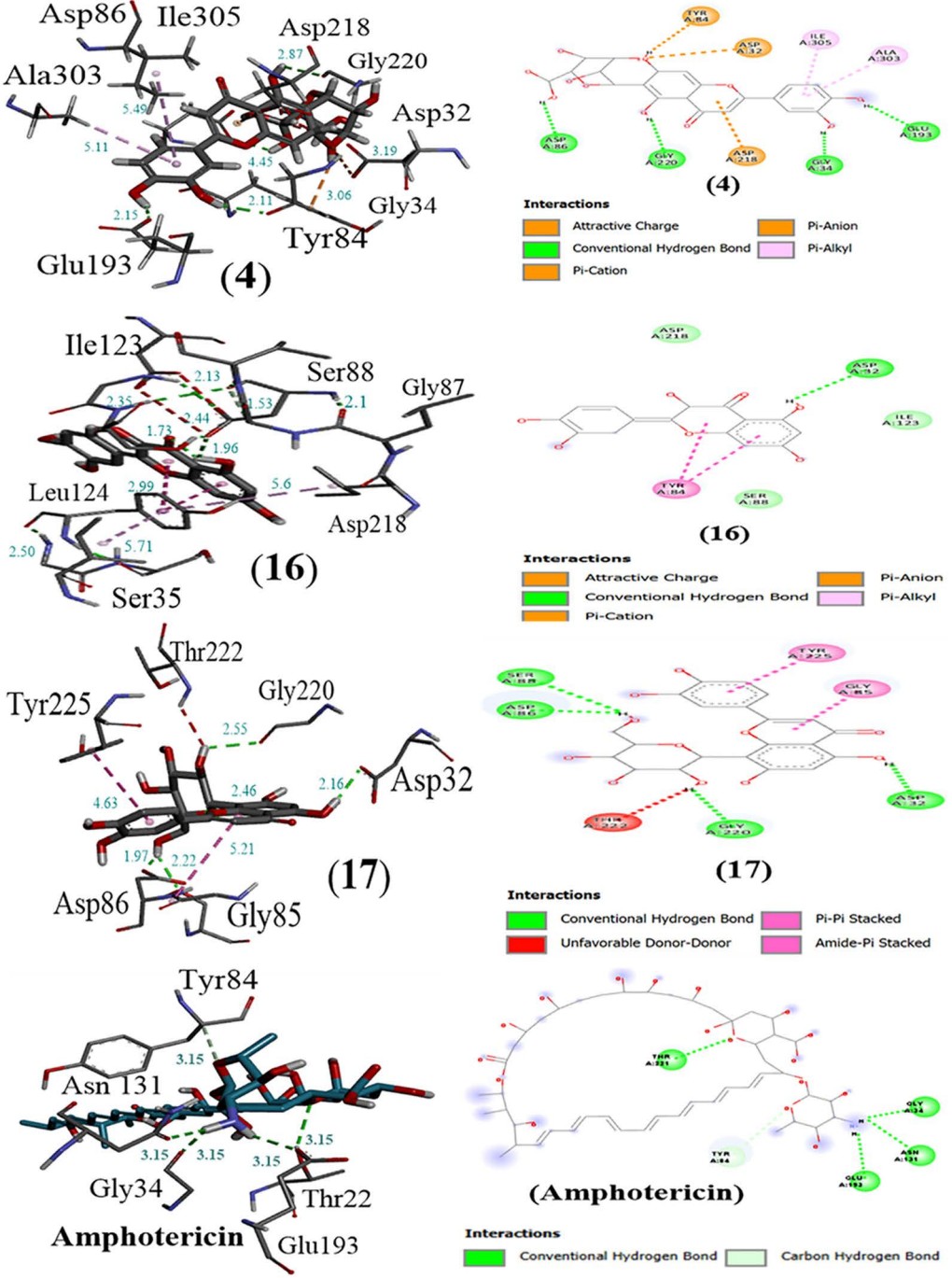

**Fig 6. The 3D and 2D interaction plot of identified bioactive compounds (4), (16), (17), and amphotericin (standard) with secreted aspartic proteinase (SAP2) protein.**

runs. The combined in vitro and in silico analysis established a mechanistic relationship between the antioxidant properties of flavonoids, their antimicrobial efficacy, and enzyme inhibition potential. The phytochemicals exhibiting stronger binding affinities also produced large inhibition zones, affirming the reliability of experimental results.

## Frontier molecular orbital (FMO) analysis

The frontier molecular orbital (FMO) analysis provides insights into the optical, electronic properties, and chemical stability of identified molecules [65]. Generally, FMOs comprise HOMO (highest occupied molecular orbital) and LUMO (lowest unoccupied molecular orbital). However, the HOMO reflects the capability to contribute an electron, while the LUMO represents the capacity to accept an electron [66,67]. The HOMO–LUMO energy gaps (Egaps) serve as critical parameters to describe the chemical hardness, chemical reactivity, chemical softness, biological activity, and dynamic stability of bioactive constituents. The compounds exhibiting larger Egap values are considered strongly resistant to electronic configuration and comprise chemically hard properties in nature. In contrast, the compounds with a small energy gap value (Egap) are highly polarizable, more reactive, softer, and display superior electronic properties. The computed HOMO, LUMO energies, Energy gaps, and IP values of structure-based compounds are summarized in Table 4 and illustrated in Fig 7. The ionization potential (IP) values of all computed compounds were compared with the ionization potential of phenol (reference, 8.33 eV) [68]. As shown in Table 4, **C4**, **C16**, and **C17** exhibited comparable energy gaps (Egaps) values. Among the studied compounds, the **C12** molecule displayed the prime Egap (6.61 eV), value with EHOMO (−6.68 eV) and ELUMO (−0.07 eV), indicating maximum electronic stability. In contrast, **C16** exhibited the minimum Egap (0.42 eV), with EHOMO of −3.81 eV and ELUMO of −3.39 eV, reflecting lower stability and higher reactivity. Furthermore, compounds **C3**, **C8**, **C11**, **C12**, and **C15,** with energy gaps 4.18, 5.01, 6.36, 6.61, and 6.58 eV, respectively, display prime values with maximum stability. The molecules **C4** (isoorientin), **C16** (quercetin), and **C17** (orientin) demonstrated their lower energy gaps, 2.50, 0.42, and 2.59 eV, respectively. The phytochemicals (**C4**, **C16,** and **C17**) experienced higher electron transfer mechanisms and exhibited higher antimicrobial characteristics with the maximum EHOMO energy, minimum energy gaps, and lower ionization potential (IP) than phenol (reference, 8.33 eV). Such electronic features are consistent with their pronounced antimicrobial activity.

The molecular electrostatic potential (MEP) surfaces shown in Fig 8 demonstrate the distribution of both positive and negative charges of constituents found in *Vitex negundo* seeds. The MEP analysis is a valuable parameter for enhancing electrophilic/nucleophilic behavior and the reactivity of diverse metabolites [69]. The MEP is a reliable tool for understanding the chemical and physical properties of molecules, including chemical reactivity, dipole moment, and partial charge distribution. In MEP maps, region of red color intensity of molecules is generally associated with negative potential, blue regions indicate positive potential, and light-colored regions indicate a neutral charge [70].

Collectively, the compounds C4, C16, and C17 exhibited more pronounced negative potential regions, indicating greater electron density and higher chemical reactivity. Ionization potential (IP) and FMO analyses further confirmed that

**Table 4. The computed HOMO energies, LUMO energies, Energy gaps, and IP(-E$_{HOMO}$) of bioactive compounds in electron volts.**

| Compounds # in Table 1 | E$_{HOMO-1}$ | E$_{HOMO}$ | E$_{LUMO}$ | E$_{LUMO+1}$ | E$_{gaps}$ | IP |
|---|---|---|---|---|---|---|
| C3 | −6.31 | −4.93 | −0.75 | −0.24 | 4.18 | 4.93 |
| C4 | −6.07 | −5.74 | −3.24 | −2.31 | 2.50 | 5.74 |
| C5 | −6.27 | −5.85 | −1.64 | −0.91 | 4.21 | 5.85 |
| C6 | −6.88 | −6.20 | −3.27 | −2.47 | 2.93 | 6.20 |
| C8 | −7.45 | −6.75 | −1.74 | −0.42 | 5.01 | 6.75 |
| C10 | −6.79 | −6.56 | −0.09 | 0.39 | 6.47 | 6.56 |
| C11 | −6.81 | −6.56 | −0.20 | 0.41 | 6.36 | 6.56 |
| C12 | −6.94 | −6.68 | −0.07 | 0.22 | 6.61 | 6.68 |
| C14 | −5.12 | −4.66 | −1.80 | −1.64 | 2.86 | 4.66 |
| C15 | −6.87 | −6.58 | −1.99 | −0.45 | 4.59 | 6.58 |
| C16 | −5.37 | −3.81 | −3.39 | −2.57 | 0.42 | 3.81 |
| C17 | −7.14 | −6.16 | −3.57 | −1.93 | 2.59 | 6.16 |

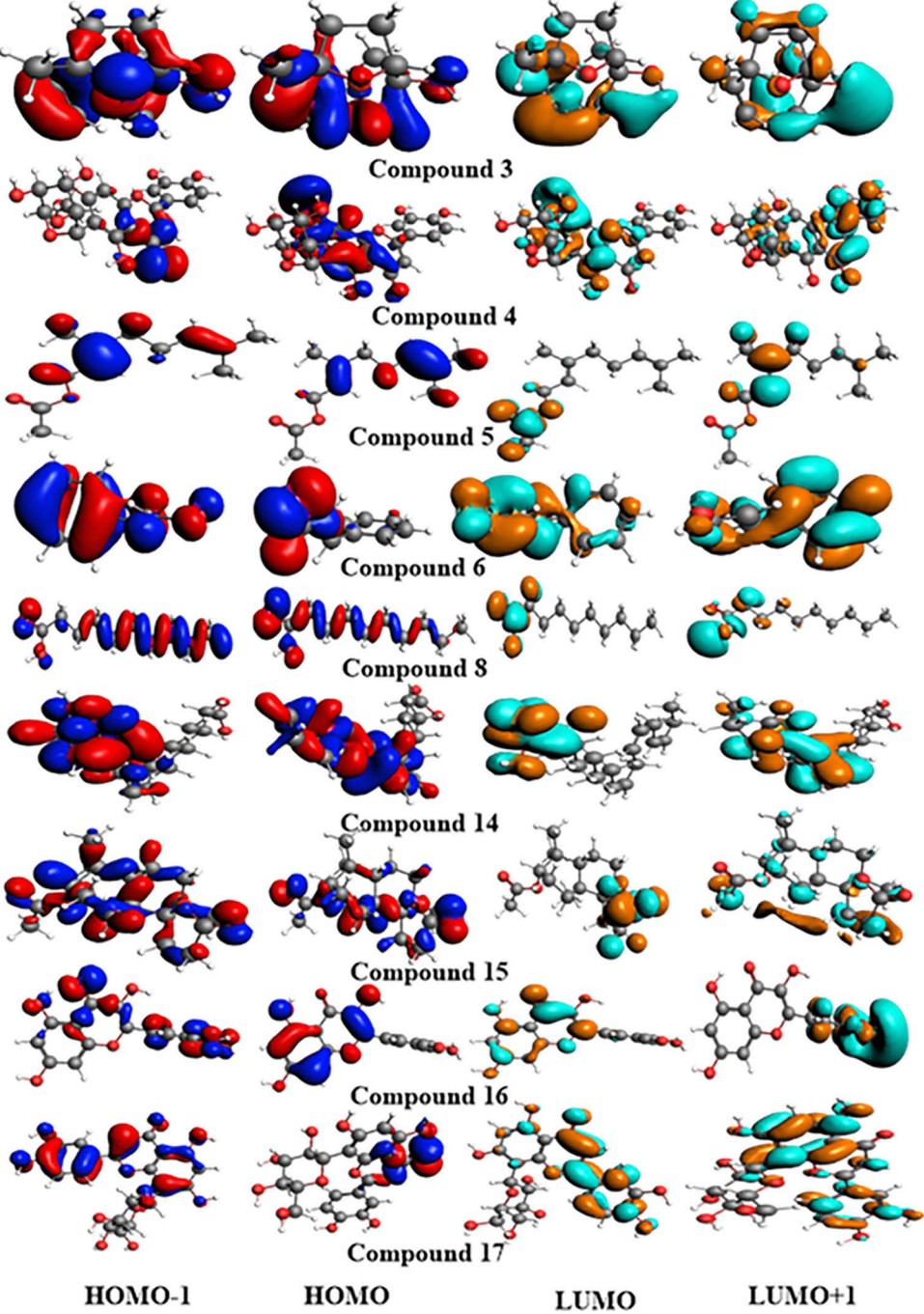

**Fig 7. The FMOs charge density distribution of the most active compounds.**

these compounds (**C4**, **C16**, and **C17**), possessing lower ionization potential and narrow energy gaps, exhibited enhanced biological reactivity and higher binding affinities with target proteins. Conversely, molecules (C10 = 6.47 eV, C11 = 6.36 eV, C12 = 6.61 eV) with larger energy gaps exhibit lower inhibitory potential, reduced chemical reactivity, greater thermodynamic stability, and slower biochemical interactions.

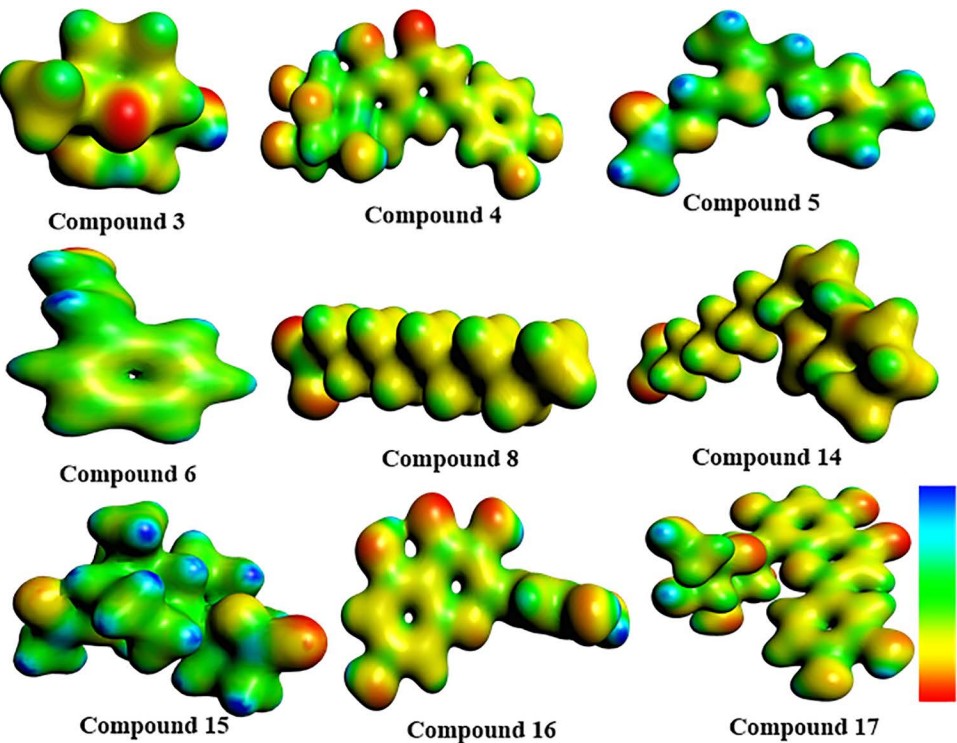

**Fig 8. MEP designed the surface outline of identified phytochemicals.** That isoorientin (**4**), quercetin (**16**), and orientin (**17**) exhibit a superior electron transfer mechanism compared to other phytochemicals displaying maximum softness, minimum hardness, and increased chemical reactivity. The equations (1 to 7) are used for the measurement of global reactivity parameters.

## Global reactivity parameters (GRPs)

The stability of FMOs (ELUMO–EHOMO = Egaps) provides insights into molecular polarizability and global reactivity parameters including global softness (S), global hardness (η), electron affinity (EA), global electrophilicity index (ω), electronegativity (X), ionization potential (IP), and the chemical potential (μ) as summarized in Table 5. An inverse relationship exists between chemical potential, hardness, stability, and energy gaps towards chemical reactivity. Consequently, phytochemicals with a higher energy gap will indicate more resistance, higher kinetic stability, and lower chemical reactivity. Table 5 presents the global hardness and chemical reactivity values of identified phytochemicals. Compound (**C4**) exhibited a hardness of 1.25 with a chemical reactivity of –4.49, while **C16** has shown a hardness of 0.21 with a chemical potential of –3.60. The phytochemical (**C17**) has a hardness of 0.65 with a chemical reactivity of –4.86. Among all molecules, the most stable phytochemical **C12** was identified after computation, shown in Table 5, with a chemical reactivity of –3.32 and a hardness of 3.23. Based on EHOMO-ELUMO gaps, the decreasing order of global hardness of all investigated molecules is graded as follows: **C12 > C10 > C11 > C8 > C15 > C5 > C3 > C6 > C14 > C4 > C17 > C16**. Next, we estimated the global softness of constituents, which is directly related to their chemical reactivity. The molecules **C10**, **C11,** and **C12** exhibited similar global softness values of 0.15, whereas **C4**, **C16,** and **C17** showed higher softness values of 0.40, 2.38, and 0.76, respectively. The decreasing trend of global softness among the investigated molecules follows the inverse order of their increasing energy gaps: **C16 > C17 > C4 > C14 > C6 > C3 > C5 > C15 > C8 > C11 > C10 > C12**. The combined analysis of global reactivity parameters, antimicrobial activity, and molecular docking indicates

Electron affinity (EA) and ionization potential (IP) values are measured using the following equations (1) and (2), respectively.

**Table 5. Global reactivity parameters calculated by using HOMO and LUMO energies.**

| Compounds# in Table 1 | IP | EA | X | η | μ | Ω | S |
|---|---|---|---|---|---|---|---|
| C3 | 4.93 | 0.75 | 2.84 | 2.09 | −2.84 | 1.92 | 0.23 |
| C4 | 5.74 | 3.24 | 4.49 | 1.25 | −4.49 | 8.06 | 0.40 |
| C5 | 5.85 | 1.64 | 3.74 | 2.10 | −3.74 | 3.32 | 0.23 |
| C6 | 6.20 | 3.27 | 4.73 | 1.46 | −4.73 | 7.66 | 0.34 |
| C8 | 6.75 | 1.74 | 4.24 | 2.50 | −4.24 | 3.59 | 0.20 |
| C10 | 6.56 | 0.09 | 3.32 | 3.23 | −3.32 | 1.70 | 0.15 |
| C11 | 6.56 | 0.20 | 3.38 | 3.18 | −3.38 | 1.79 | 0.15 |
| C12 | 6.68 | 0.07 | 3.37 | 3.30 | −3.37 | 1.71 | 0.15 |
| C14 | 4.66 | 1.80 | 3.23 | 1.43 | −3.23 | 3.64 | 0.34 |
| C15 | 6.58 | 1.99 | 4.28 | 2.29 | −4.28 | 3.99 | 0.21 |
| C16 | 3.81 | 3.39 | 3.60 | 0.21 | −3.60 | 30.85 | 2.38 |
| C17 | 6.16 | 3.57 | 4.86 | 0.65 | −4.86 | 18.16 | 0.76 |

IP = ionization potential, ω = global electrophilicity, EA = electron affinity, x = electronegativity, η = global hardness, μ = chemical potential, S = global softness, all in eV units.

$$EA = -E_{LUMO} \tag{1}$$

$$IP = -E_{HOMO} \tag{2}$$

Hardness (η) and Electronegativity (X) values are attained by equations (3) and (4), respectively.

$$\eta = \frac{IP - EA}{2} \tag{3}$$

$$X = \frac{IP + EA}{2} \tag{4}$$

To calculate the chemical potential ((μ), equation (5) is used

$$\mu = \frac{E_{HOMO} + E_{LUMO}}{2} \tag{5}$$

The electrophilicity (ω) is calculated by using equation (6).

$$\omega = \frac{\mu^2}{2\eta} \tag{6}$$

For softness (S), the calculation equation (7) is used.

$$\sigma = \frac{1}{2\eta} \tag{7}$$

## TD-DFT analysis and biological implications

The compounds **C4**, **C16**, and **C17** were computed using TD-DFT at the B3LYP/6-311G level of study. During computation analysis, only six (singlet to singlet) transitions were calculated during the computational analysis. The computed λ max values ranged between 1541 and 285 nm, corresponding to transition energies from 0.80 to 4.35 eV, indicating electronic transition in both the UV and visible regions. The calculated UV–visible spectral parameters, including transition energies (eV), oscillator strengths ($f_{os}$), wavelength (nm), transition moment ($M_x^{gm}$ a.u.), light harvesting efficiency (LHE), and molecular orbital transition assignments are summarized in Table 6. The corresponding absorption spectra of the investigated molecules are presented in Fig 9.

Compound (**4**) exhibited a low transition energy (0.8045 eV) with a long wavelength of 1541.16 nm at a relatively weak oscillator strength (0.0270). The harvesting value for its transition states $S_0 \rightarrow S_1$ (0.061) and $S_0 \rightarrow S_2$ (0.395) indicated a moderate light-harvesting potential. In contrast, C16 has shown the highest transition energy, 3.8629 eV for $S_0 \rightarrow S_3$ at 320.96 nm as compared to C4. The compound C17 showed a high transition moment (0.6617 a.u.), suggesting efficient light harvesting property (0.6192) at wavelength 332.63 nm for $S_0 \rightarrow S_1$ among the studied molecules. From the studied compounds, C17 displayed the most promising light-harvesting applications, followed by C4, with moderate potential, whereas C16 exhibited comparatively lower efficiency.

The optical performance of the investigated compounds was further assessed using the light harvesting efficiency (LHE) parameter [71]. A nonlinear optical (NLO) response of a molecule is commonly associated with larger LHE values. The light-harvesting efficiency of the studied molecules was calculated through Equation (8), and the results are summarized in Table 6 [72,73].

$$LHE = 1 - 10^{-f} \tag{8}$$

The optical properties of molecules serve as indicative parameters of enhanced redox flexibility, which may facilitate interactions with bacterial redox enzymes and proteins. Molecules with lower transition energies and higher oscillator strengths ($f_{os}$) commonly exhibit greater bioactivity due to stabilization of enzyme–ligand complexes and efficient charge transfer mechanism. The combined DFT and molecular docking analyses revealed that C4 (isoorientin), C16 (quercetin), and C17 (orientin) possess the most favorable electronic and physicochemical attributes for antimicrobial activity. Their reduced energy gaps, strong binding affinities, and effective charge-transfer potential suggest that these compounds act as strong inhibitors, stabilized through both covalent and non-covalent interactions. The experimental inhibition data showed excellent correlation with theoretical predictions, indicating a strong relationship between electronic reactivity and antimicrobial efficacy.

**Table 6. The computed transition energies (eV), oscillator strengths ($f_{os}$), wavelengths (nm), transition moments ($M_x^{gm}$ a.u.), light harvesting efficiency (LHE), and orbital transition states.**

| Compound | Transition state | $E$(eV) | $\lambda_{max}$ (nm) | $f_{os}$ | $M_x^{gm}$(a.u.) | LHE |
|---|---|---|---|---|---|---|
| C4 | $S_0 \rightarrow S_1$ | 0.8045 | 1541.16 | 0.0270 | 0.692 (95.85%) | 0.061 |
| | $S_0 \rightarrow S_2$ | 1.7350 | 714.62 | 0.2179 | 0.1568 (4.92%) | 0.395 |
| | $S_0 \rightarrow S_3$ | 2.0869 | 594.11 | 0.0913 | 0.1376 (3.78%) | 0.190 |
| C16 | $S_0 \rightarrow S_1$ | 3.8629 | 320.96 | 0.1426 | 0.687 (94.44%) | 0.279 |
| | $S_0 \rightarrow S_2$ | 4.1455 | 299.08 | 0.0010 | 0.15890 (5.0%) | 0.003 |
| | $S_0 \rightarrow S_3$ | 4.3502 | 285.01 | 0.0001 | 0.661 (87.56%) | 0.001 |
| C17 | $S_0 \rightarrow S_1$ | 3.7274 | 332.63 | 0.4193 | −0.14880 (4.4%) | 0.619 |
| | $S_0 \rightarrow S_2$ | 3.9543 | 313.54 | 0.0212 | 0.676 (91.53%) | 0.047 |
| | $S_0 \rightarrow S_3$ | 4.1569 | 298.26 | 0.0102 | 0.673 (90.72%) | 0.024 |

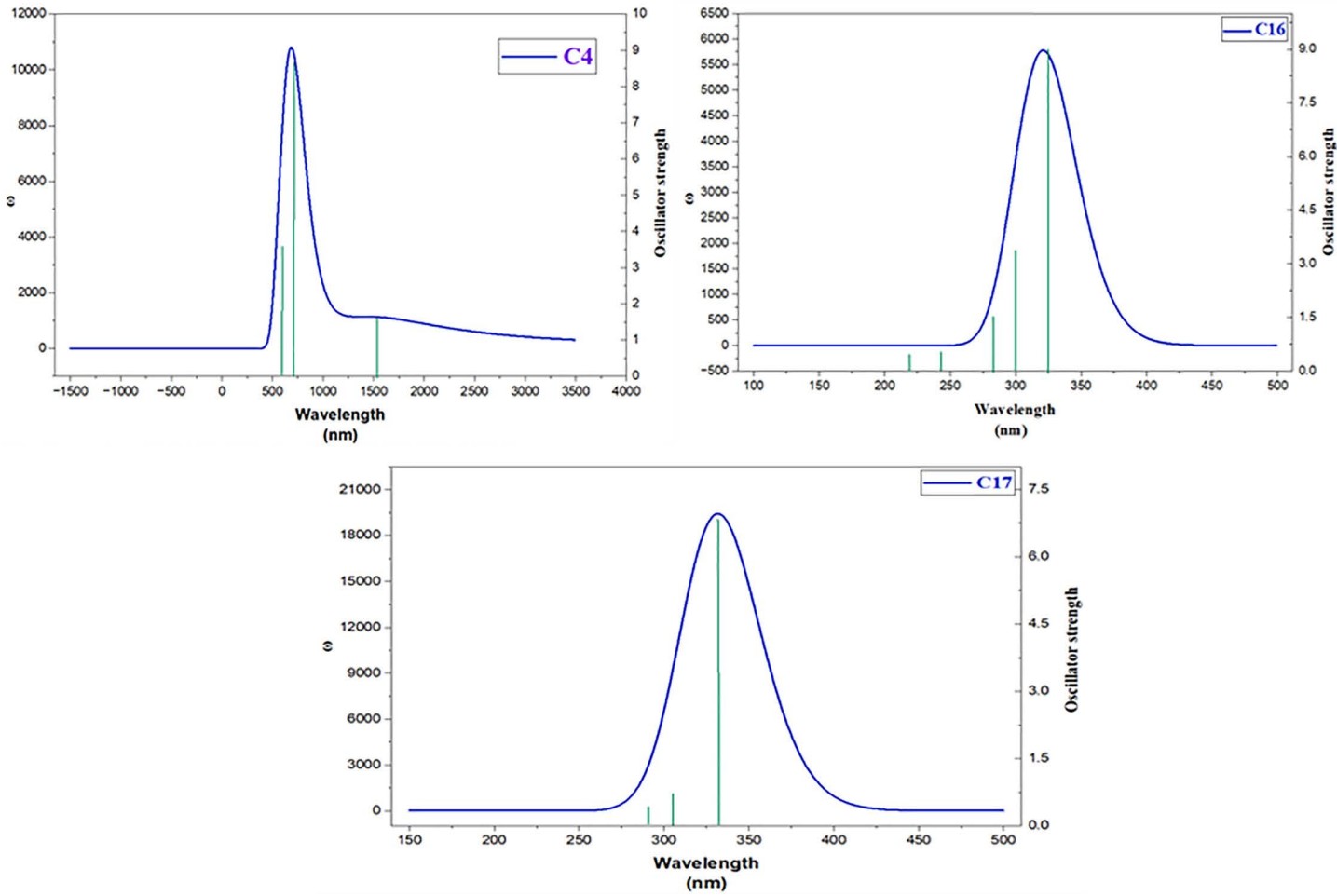

**Fig 9. UV-visible spectra of the most active compounds (C4, C16, and C17).**

## Conclusions

This integrated study employed UHPLC-QTOF-MS/MS-based metabolic profiling, density functional theory (DFT) calculations, and molecular docking simulations to characterize the molecular composition and electronic behaviors of *Vitex negundo* seed constituents. Among the seventeen compounds, the flavonoids such as isoorientin (**C4**), quercetin (**C16**), and orientin (**C17**) are reported for the first time with MS/MS spectra, molecular docking, and TDDFT analyses against bacterial and fungal targets, supported by chemical reactivity and FMO parameters. Quercetin (**C16**) exhibited the highest binding energy of −8.4 kcal/mol against the secreted aspartic proteinase (SAP2) and −7.8 kcal/mol energy against the Staph Gyrase B, indicating strongest inhibition potential among investigated phytochemicals. The methanolic seed extract showed significant antibacterial and antifungal activities with inhibition zone of 25.7±0.4 (against *Candida albicans*) and 26.4±0.3 (against *Staphylococcus aureus*). The global reactivity parameters revealed the following decreasing trend in softness: **C16 > C17 > C4 > C14 > C6 > C3 > C5 > C15 > C8 > C11 > C10 > C12**, inversely correlated with molecular hardness and energy band gaps. The identification of these bioactive compounds provides promising molecular templates and chemical leads against multidrug-resistant microbes. While previous studies have employed spectroscopic and chromatographic techniques to investigate *V. negundo* seeds, the present work uniquely integrates UHPLC-QTOF-MS/

MS with molecular docking and DFT analyses to elucidate their antimicrobial potential. These findings offer the mechanistic insights into the antimicrobial behavior of *V. negundo* seeds; however, further pharmacological and in vivo studies are essential to confirm their therapeutic efficacy.

## Supporting information

**S1 File.** S1. Antibacterial assay; **S2**. Antifungal assay. **S3**. UHPLC-QTOF-MS/MS-based metabolic profiling of *Vitex negundo* seeds in positive ionization mode procedure; **S4** – **S8**. Description of **Compound 11** – **Compound 15**; **S9**. Fig 5 and Fig 6. Docking simulations of major bioactive constituents.
(DOCX)

**S1 Data. Raw data – 1.1 Docking.**
(ZIP)

**S2 Data. Raw data – 1.2 Docking.**
(ZIP)

**S3 Data. Raw data – 2 Docking.**
(ZIP)

**S4 Data. Raw data – 2.1 – Docking.**
(ZIP)

**S5 Data. Raw data – 3 DFT.**
(ZIP)

**S6 Data. Raw data – 3.1.1.**
(ZIP)

**S7 Data. Raw data – 3.1.**
(ZIP)

**S8 Data. Raw data – 3.2 DFT.**
(ZIP)

**S9 Data. Raw data – 3.3.1DFT.**
(ZIP)

**S10 Data. Raw data – 3.3.2DFT.**
(ZIP)

**S11 Data. Raw data – 3.3DFT.**
(ZIP)

**S12 Data. Raw data – 3.4DFT.**
(ZIP)

**S13 Data. Raw data 1.0 Docking.**
(ZIP)

**S14 Data. Raw data, file 3.4.1 DFT.**
(ZIP)

**S15 Data. Raw data, file 3.4.2 DFT.**
(ZIP)

**S16 Data. Raw data, file 4.0.**
(ZIP)

## Author contributions

**Conceptualization:** Javed Mustafa, Tuba Ashraf.

**Methodology:** Javed Mustafa, Muhammad Yunis.

**Resources:** Shazia Kousar, Usman Rahim.

**Software:** Basharat Ali.

**Supervision:** Bakhat Ali, Muhammad Imran.

**Writing – original draft:** Javed Mustafa.

**Writing – review & editing:** Shazia Kousar, Adeem Mahmood, Saif Ullah.

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
