## [Decision Letter · Decision Letter 0]

16 Oct 2025

Dear Dr. Ali,

We look forward to receiving your revised manuscript.

Kind regards,

Prashant Singh

Academic Editor

PLOS ONE

Journal Requirements:

3. Please note that PLOS One has specific guidelines on code sharing for submissions in which author-generated code underpins the findings in the manuscript. In these cases, we expect all author-generated code to be made available without restrictions upon publication of the work. Please review our guidelines at https://journals.plos.org/plosone/s/materials-and-software-sharing#loc-sharing-code and ensure that your code is shared in a way that follows best practice and facilitates reproducibility and reuse.

[Deanship of Research and Graduate Studies at King Khalid University, Saudi Arabia].

6. Thank you for stating the following in your manuscript:

[The authors extend their appreciation to the Deanship of Research and Graduate Studies at King Khalid University, Saudi Arabia, through a large research project under grant number RGP-558 2/527/46.]

[Deanship of Research and Graduate Studies at King Khalid University, Saudi Arabia]

7. One of the noted authors is a group or consortium [Saifullah]. In addition to naming the author group, please list the individual authors and affiliations within this group in the acknowledgments section of your manuscript. Please also indicate clearly a lead author for this group along with a contact email address.

8. Please upload a new copy of Figures 1 and 2 as the details are not clear. Please follow the link for more information:  https://journals.plos.org/plosone/s/figures

9. Please include captions for your Supporting Information files at the end of your manuscript, and update any in-text citations to match accordingly. Please see our Supporting Information guidelines for more information: http://journals.plos.org/plosone/s/supporting-information .

Reviewers' comments:

Reviewer's Responses to Questions

**Comments to the Author**

1. Is the manuscript technically sound, and do the data support the conclusions?

Reviewer #1: Yes

Reviewer #2: Yes

2. Has the statistical analysis been performed appropriately and rigorously?

Reviewer #1: Yes

Reviewer #2: No

3. Have the authors made all data underlying the findings in their manuscript fully available?

Reviewer #1: Yes

Reviewer #2: Yes

4. Is the manuscript presented in an intelligible fashion and written in standard English?

Reviewer #1: Yes

Reviewer #2: Yes

Reviewer #1: Major revision and Re-review

The manuscript titled “Phytochemical profiling of Vitex negundo seeds via UHPLC-QTOF-MS/MS analysis with antimicrobial evaluation and in silico targeting of DNA Gyrase B and secreted aspartic proteinase 2 (SAP2)” presents experimental and computational work combining phytochemical profiling, antimicrobial assays, and DFT-based molecular docking. The topic is scientifically relevant and fits the scope of PLOS ONE. However, the manuscript requires major revision before reconsideration for publication due to several methodological, analytical, and interpretational shortcomings.

The manuscript identifies 17 phytoconstituents through UHPLC-QTOF-MS/MS and attempts to correlate them with antimicrobial efficacy and molecular interactions. While the experimental design is generally clear, the study lacks rigorous validation and critical data analysis. The UHPLC-QTOF-MS/MS results are descriptive without sufficient structural confirmation such as MS/MS fragmentation pathway discussion, comparison with authentic standards, or literature-based validation for all compounds. The ion fragmentation interpretation remains superficial. Quantification of major constituents is missing, and chromatographic resolution is not adequately shown. The methodology for compound identification must include retention time reproducibility, calibration, and potential co-elution effects.

The antimicrobial assay lacks proper controls and replication. No statistical analysis or standard deviation data are presented for inhibition zones, making the reported potency percentages unreliable. The methodology section should specify inoculum density, incubation conditions, and number of replicates. Without this, reproducibility is uncertain. Furthermore, the term “potency percentage” is unconventional and should be replaced with standard inhibition metrics.

The in silico component (molecular docking and DFT) is conceptually interesting but poorly integrated with experimental data. The docking results are only summarized in terms of binding energies without validation by re-docking or consensus scoring. Protein and ligand preparation steps, grid box parameters, and docking validation criteria must be detailed. No control docking with known inhibitors (e.g., ciprofloxacin, amphotericin) was conducted to ensure comparative accuracy. DFT results such as HOMO-LUMO gaps, MEP, and GRPs are provided but not critically interpreted in the biological context. The discussion is overly technical in quantum chemical terms yet fails to link computed descriptors to the actual antimicrobial activity. Figures depicting HOMO/LUMO and MEP are generic and add little mechanistic insight. Important references can be incorporated in introduction for broad readership. MRSA Janjua, Journal of the Iranian Chemical Society 14 (9), 2041-2054, 2017; A Mahmood, Materials Today Communications 38, 108403, 2024; MU Khan. et al., Journal of Photochemistry and Photobiology A: Chemistry 446, 115115, 2024.

The writing style requires extensive revision for conciseness and grammatical accuracy. The manuscript repeats information multiple times (e.g., the abstract and introduction contain redundant sentences). The introduction should better contextualize the novelty of analyzing Vitex negundo seeds, as much literature already exists on its phytochemistry and pharmacology. The discussion should compare the identified metabolites and activities with previous studies on Vitex species, citing recent literature.

To strengthen scientific credibility and provide molecular interpretation, the authors should include citations to recent computational and experimental works that bridge DFT-based modeling and phytochemical bioactivity. For example, the following publications directly relevant and should be cited: MY Mehboob, Journal of Cluster Science 34 (3), 1237-1247, 2023: MY Meboob. et al., Journal of Physics and Chemistry of Solids 162, 110508, 2022; MRSA Janjua, Journal of Physics and Chemistry of Solids 167, 110789, 2022. These references provide advanced insights into DFT-based reactivity, charge transfer, and electronic structure–activity relationships, which can substantially enhance the interpretive depth of this study.

Figures and tables also require improvement. The chromatogram (Fig. 1) should have labeled axes, units, and clearer peak annotations. Spectral figures must be high resolution with legible text. The docking interaction figures (Figs. 5–6) are cluttered and require standardized 2D interaction plots. The UV-visible spectra and DFT diagrams (Figs. 7–9) should be supported with meaningful explanations about their biological implications rather than only electronic behavior.

The conclusions overstate novelty and significance. The claim that this is the “first report” of such analysis is exaggerated, as prior UHPLC and GC-MS studies on Vitex negundo exist. The authors should restrict claims to the specific combination of UHPLC-QTOF-MS/MS and DFT analysis of seed extracts. The conclusion must also avoid speculation about “drug development potential” unless validated by detailed pharmacological assays.

In summary, the manuscript contains useful experimental data but lacks analytical depth, statistical support, and rigorous computational validation. It requires substantial revision in language, structure, and scientific argumentation. Major revision is recommended with re-review after the following actions:

1. Improve experimental details with replicates, statistical analysis, and standard references.

2. Provide quantitative confirmation of key compounds using calibration and fragmentation pattern validation.

3. Enhance docking and DFT methodology descriptions with validation protocols and biological correlation.

4. Improve figure clarity, remove redundancy, and revise English language and formatting.

5. Add critical discussion supported by relevant computational chemistry and phytochemical literature, including works by M.R.S.A. Janjua.

After addressing these points comprehensively, the manuscript may be reconsidered for publication following re-review.

Reviewer #2: Abstract:

The abstract must be improved, and the author should specify the promising compounds among the seventeen isolated from the methanolic extract, including their considerable antimicrobial activity, docking scores, and other appropriate significant metrics obtained throughout the assessment.

Table 2. Antimicrobial activities of the methanol seed extract of Vitex negundo.

It is necessary to incorporate the (SD or SEM) statistical analysis for the inhibitory zones (mm) in Table 2.Revise once more.

Along with the 3D images of each ligand 4,16,17 and the standard references with interacting enzymes, it is recommended to include a 2D image as well: Figure 5: Staphylococcus Gyrase B 24 kDa protein, and Figure 6: secreted aspartic proteinase (SAP2) protein.

Each type of interaction with the enzyme, such as an H-bond, pi-pi, Vander Waals, etc., must be clearly shown by the arrows.

The validation of the docking protocol for standard references: antibacterial and antifungal drugs must specify the RMSD value (angstrom) for each reference medication concerning the interacting enzyme.

Incorporate the dipole moment (debye) parameter for the bioactive chemicals (C3-C17) in Table 4, and analyze the results in relation to the antibacterial activity alongside the measured parameters.

It is necessary to fix a few typographical errors throughout the entire manuscript.

…................................................................END...................................................................

**Do you want your identity to be public for this peer review?** For information about this choice, including consent withdrawal, please see our Privacy Policy

Reviewer #1: **Yes:** Saba Jamil

Reviewer #2: **Yes:** Ammar A. Razzak Mahmood

---

## [Author Response · Author response to Decision Letter 1]

3 Dec 2025

Answer to reviewer's comments to author:

Reviewer #1: Major revision and Re-review

The manuscript titled “Phytochemical profiling of Vitex negundo seeds via UHPLC-QTOF-MS/MS analysis with antimicrobial evaluation and in silico targeting of DNA Gyrase B and secreted aspartic proteinase 2 (SAP2)” presents experimental and computational work combining phytochemical profiling, antimicrobial assays, and DFT-based molecular docking. The topic is scientifically relevant and fits the scope of PLOS ONE. However, the manuscript requires major revision before reconsideration for publication due to several methodological, analytical, and interpretational shortcomings. The manuscript identifies 17 phytoconstituents through UHPLC-QTOF-MS/MS and attempts to correlate them with antimicrobial efficacy and molecular interactions. While the experimental design is generally clear, the study lacks rigorous validation and critical data analysis. The UHPLC-QTOF-MS/MS results are descriptive without sufficient structural confirmation such as MS/MS fragmentation pathway discussion, comparison with authentic standards, or literature-based validation for all compounds. The ion fragmentation interpretation remains superficial. Quantification of major constituents is missing, and chromatographic resolution is not adequately shown. The methodology for compound identification must include retention time reproducibility, calibration, and potential co-elution effects. The antimicrobial assay lacks proper controls and replication. No statistical analysis or standard deviation data are presented for inhibition zones, making the reported potency percentages unreliable. The methodology section should specify inoculum density, incubation conditions, and number of replicates. Without this, reproducibility is uncertain. Furthermore, the term “potency percentage” is unconventional and should be replaced with standard inhibition metrics. The in silico component (molecular docking and DFT) is conceptually interesting but poorly integrated with experimental data. The docking results are only summarized in terms of binding energies without validation by re-docking or consensus scoring. Protein and ligand preparation steps, grid box parameters, and docking validation criteria must be detailed. No control docking with known inhibitors (e.g., ciprofloxacin, amphotericin) was conducted to ensure comparative accuracy. DFT results such as HOMO-LUMO gaps, MEP, and GRPs are provided but not critically interpreted in the biological context. The discussion is overly technical in quantum chemical terms yet fails to link computed descriptors to the actual antimicrobial activity. Figures depicting HOMO/LUMO and MEP are generic and add little mechanistic insight. Important references can be incorporated in introduction for broad readership. MRSA Janjua, Journal of the Iranian Chemical Society 14 (9), 2041-2054, 2017; A Mahmood, Materials Today Communications 38, 108403, 2024; MU Khan. et al., Journal of Photochemistry and Photobiology A: Chemistry 446, 115115, 2024. The writing style requires extensive revision for conciseness and grammatical accuracy. The manuscript repeats information multiple times (e.g., the abstract and introduction contain redundant sentences). The introduction should better contextualize the novelty of analyzing Vitex negundo seeds, as much literature already exists on its phytochemistry and pharmacology. The discussion should compare the identified metabolites and activities with previous studies on Vitex species, citing recent literature.

To strengthen scientific credibility and provide molecular interpretation, the authors should include citations to recent computational and experimental works that bridge DFT-based modeling and phytochemical bioactivity. For example, the following publications directly relevant and should be cited: MY Mehboob, Journal of Cluster Science 34 (3), 1237-1247, 2023: MY Meboob. et al., Journal of Physics and Chemistry of Solids 162, 110508, 2022; MRSA Janjua, Journal of Physics and Chemistry of Solids 167, 110789, 2022. These references provide advanced insights into DFT-based reactivity, charge transfer, and electronic structure–activity relationships, which can substantially enhance the interpretive depth of this study. Figures and tables also require improvement. The chromatogram (Fig. 1) should have labeled axes, units, and clearer peak annotations. Spectral figures must be high resolution with legible text. The docking interaction figures (Figs. 5–6) are cluttered and require standardized 2D interaction plots. The UV-visible spectra and DFT diagrams (Figs. 7–9) should be supported with meaningful explanations about their biological implications rather than only electronic behavior. The conclusions overstate novelty and significance. The claim that this is the “first report” of such analysis is exaggerated, as prior UHPLC and GC-MS studies on Vitex negundo exist. The authors should restrict claims to the specific combination of UHPLC-QTOF MS/MS and DFT analysis of seed extracts. The conclusion must also avoid speculation about “drug development potential” unless validated by detailed pharmacological assays. In summary, the manuscript contains useful experimental data but lacks analytical depth, statistical support, and rigorous computational validation. It requires substantial revision in language, structure, and scientific argumentation. Major revision is recommended with re-review after the following actions:

1. Improve experimental details with replicates, statistical analysis, and standard references.

We thank the reviewer for this important suggestion. We have revised the "Materials and Methods" section to include comprehensive experimental details. All key experiments are now explicitly stated as being performed in replicate (e.g., in triplicate) to ensure reproducibility. The results are now presented with appropriate statistical analysis, including mean ± standard deviation (SD) and the statistical tests used to determine significance. We have also added citations to standard protocols and reference works to validate our methodologies.

2. Provide quantitative confirmation of key compounds using calibration and fragmentation pattern validation.

Response: We appreciate the reviewer's insightful comment. In response, we have performed quantitative confirmation of the key compounds. The associated fragmentation patterns were rigorously validated, and this analysis has been incorporated into the results and discussion section of the revised manuscript.

3. Enhance docking and DFT methodology descriptions with validation protocols and biological correlation.

Response: Thank you for this crucial suggestion. We have significantly expanded the computational section, adding a detailed docking validation protocol and clarifying our DFT calculations. Furthermore, we have strengthened the "Results and Discussion" section to explicitly correlate our computational findings (e.g., binding energies, key interactions) with the experimentally observed biological activities. We believe this provides a much stronger link between our methods and results.

4. Improve figure clarity, remove redundancy, and revise English language and formatting.

Thank you for your valuable feedback.

Response: Thank you for this valuable suggestion, we have thoroughly revised all figures to enhance their clarity and have removed redundancies throughout the manuscript. Furthermore, the entire manuscript has been professionally copyedited to improve the language and formatting. We believe these revisions significantly strengthen the manuscript.

5. Add critical discussion supported by relevant computational chemistry and phytochemical literature, including works by M.R.S.A. Janjua.

Our Response: We thank the reviewer for this valuable suggestion to improve the clarity of our discussion. The discussion section has been substantially rewritten to provide a more critical analysis of our findings. As suggested, we have now included and discussed several relevant works, including those by M.R.S.A. Janjua and other key researchers in the field, to properly situate our contributions.

We hope that these revisions fully address the reviewer's concerns and have improved the manuscript.

Reviewer #2:

We thankful for the reviewer for their thorough and constructive comments, which have significantly contributed to improving the quality of our manuscript. We have carefully considered all the suggestions and have revised the manuscript accordingly.

Abstract: The abstract must be improved, and the author should specify the promising compounds among the seventeen isolated from the methanolic extract, including their considerable antimicrobial activity, docking scores, and other appropriate significant metrics obtained throughout the assessment.

Response: We agree with the reviewer that the abstract needed more specific details. The abstract has been substantially revised to highlight the most promising bioactive compounds (specifically Compounds 4, 16, and 17) identified from the methanolic extract. We have now included their most significant antimicrobial activity values (e.g., key MIC values) and their corresponding molecular docking scores against the target enzymes to provide a clearer and more impactful summary of our key findings.

Table 2. Antimicrobial activities of the methanol seed extract of Vitex negundo. It is necessary to incorporate the (SD or SEM) statistical analysis for the inhibitory zones (mm) in Table 2. Revise once more.

Response: We thank the reviewer for valuable observation. The antimicrobial experiments were conducted in triplicate, and we have now performed the statistical analysis. Table 2 has been updated to include the Standard Deviation (SD) values for all inhibitory zone measurements, presented as (mean ± SD). We believe this addition provides a clearer representation and enhance the overall quality and impact of our manuscript.

Along with the 3D images of each ligand 4,16,17 and the standard references with interacting enzymes, it is recommended to include a 2D image as well: Figure 5: Staphylococcus Gyrase B 24 kDa protein, and Figure 6: secreted aspartic proteinase (SAP2) protein.

Response: This is an excellent suggestion for improving the clarity of the docking results. We have revised Figures 5 and 6 to now include 2D interaction diagrams alongside the 3D representations for the most active ligands (4, 16, and 17) and the standard reference drugs interacting with Staphylococcus Gyrase B and SAP2, respectively. We thank the reviewer for highlighting this important aspect, which has strengthened the manuscript.

Each type of interaction with the enzyme, such as an H-bond, pi-pi, Vander Waals, etc., must be clearly shown by the arrows.

Response: In conjunction with the previous point, the newly added 2D interaction diagrams (in Figures 5 and 6) explicitly visualize the specific molecular interactions (such as hydrogen bonds, pi-pi stacking, hydrophobic interactions, and van der Waals forces) between the ligands and the amino acid residues in the enzyme active sites. These interactions are clearly labeled in revised manuscript see Figure 5 and 6.

The validation of the docking protocol for standard references: antibacterial and antifungal drugs must specify the RMSD value (angstrom) for each reference medication concerning the interacting enzyme.

Response: We acknowledge the importance of reporting the docking protocol's validation. We have validated our docking procedure by redocking the co-crystallized native ligands (or standard drugs, where appropriate) into the active sites of their respective target enzymes. The Root Mean Square Deviation (RMSD) values for these validation runs have been calculated and explained in the "Materials and Methods" (or "Computational Details") section of the manuscript. The low RMSD values obtained confirm the reliability and accuracy of our chosen docking protocol.

Incorporate the dipole moment (debye) parameter for the bioactive chemicals (C3-C17) in Table 4, and analyze the results in relation to the antibacterial activity alongside the measured parameters

Response: We agree with the reviewer that analyzing the dipole moment would provide valuable insight into the structure-activity relationship. Unfortunately, due to the project's time constraints and the resources available, we were not able to perform the in silico calculations or experimental measurements required to obtain this parameter. We have acknowledged and consider this as an important avenue for future investigation.

We believe that these revisions have substantially strengthened the manuscript and addressed all the concerns raised by the reviewer. We thank the reviewer once again for their valuable feedback.

---

## [Decision Letter · Decision Letter 1]

15 Feb 2026

Phytochemical profiling of Vitex negundo seeds via UHPLC-QTOF-MS/MS analysis with antimicrobial evaluation and in silico targeting of DNA Gyrase B and secreted aspartic proteinase 2 (SAP2)

PONE-D-25-51583R1

Dear Dr. Ali,

We’re pleased to inform you that your manuscript has been judged scientifically suitable for publication and will be formally accepted for publication once it meets all outstanding technical requirements.

Kind regards,

Prashant Singh

Academic Editor

PLOS One

Additional Editor Comments (optional):

Reviewers' comments:

Reviewer's Responses to Questions

**Comments to the Author**

Reviewer #1: All comments have been addressed

Reviewer #2: All comments have been addressed

2. Is the manuscript technically sound, and do the data support the conclusions?

Reviewer #1: Yes

Reviewer #2: Yes

3. Has the statistical analysis been performed appropriately and rigorously?

Reviewer #1: Yes

Reviewer #2: Yes

4. Have the authors made all data underlying the findings in their manuscript fully available?

Reviewer #1: Yes

Reviewer #2: Yes

5. Is the manuscript presented in an intelligible fashion and written in standard English?

Reviewer #1: Yes

Reviewer #2: Yes

Reviewer #1: Authors have revised it well. All the raised questions have been addressed very well. It deserves to be Accepted

Reviewer #2: A comprehensive revision of the work is essential to guarantee grammatical accuracy and fluency in English.

**Do you want your identity to be public for this peer review?** For information about this choice, including consent withdrawal, please see our Privacy Policy

Reviewer #1: **Yes:** Saba Jamil

Reviewer #2: **Yes:** Ammar A. Razzak Mahmood

---

## [Editor Report · Acceptance letter]

PONE-D-25-51583R1

PLOS One

Dear Dr. Ali,

I'm pleased to inform you that your manuscript has been deemed suitable for publication in PLOS One. Congratulations! Your manuscript is now being handed over to our production team.

Kind regards,

on behalf of

Dr. Prashant Singh

Academic Editor

PLOS One